# Inhibition of *CERS1* in skeletal muscle exacerbates age-related muscle dysfunction

**Martin Wohlwend[1]\*, Pirkka-Pekka Laurila[1], Ludger JE Goeminne[1], Tanes Lima[1], Ioanna Daskalaki[1], Xiaoxu Li[1], Giacomo von Alvensleben[1], Barbara Crisol[1], Renata Mangione[1], Hector Gallart-Ayala[2], Amélia Lalou[1], Olivier Burri[3], Stephen Butler[4], Jonathan Morris[4], Nigel Turner[5,6], Julijana Ivanisevic[2], Johan Auwerx[1]\***

[1]Laboratory of Integrative Systems Physiology, École Polytechnique Fédérale de Lausanne, Lausanne, Switzerland; [2]Metabolomics Platform, Faculty of Biology and Medicine, University of Lausanne (UNIL), Lausanne, Switzerland; [3]Bioimaging and optics platform, École polytechnique fédérale de Lausanne (EPFL), Lausanne, Switzerland; [4]School of Chemistry, University of New South Wales Sydney, Sydney, Australia; [5]Cellular Bioenergetics Laboratory, Victor Chang Cardiac Research Institute, Darlinghurst, Australia; [6]School of Biomedical Sciences, University of New South Wales Sydney, Sydney, Australia

**\*For correspondence:**
wohlwendm@gmail.com (MW);
admin.auwerx@epfl.ch (JA)

**Abstract** Age-related muscle wasting and dysfunction render the elderly population vulnerable and incapacitated, while underlying mechanisms are poorly understood. Here, we implicate the CERS1 enzyme of the de novo sphingolipid synthesis pathway in the pathogenesis of age-related skeletal muscle impairment. In humans, *CERS1* abundance declines with aging in skeletal muscle cells and, correlates with biological pathways involved in muscle function and myogenesis. Furthermore, *CERS1* is upregulated during myogenic differentiation. Pharmacological or genetic inhibition of *CERS1* in aged mice blunts myogenesis and deteriorates aged skeletal muscle mass and function, which is associated with the occurrence of morphological features typical of inflammation and fibrosis. Ablation of the *CERS1* orthologue *lagr-1* in *Caenorhabditis elegans* similarly exacerbates the age-associated decline in muscle function and integrity. We discover genetic variants reducing *CERS1* expression in human skeletal muscle and Mendelian randomization analysis in the UK biobank cohort shows that these variants reduce muscle grip strength and overall health. In summary, our findings link age-related impairments in muscle function to a reduction in *CERS1*, thereby underlining the importance of the sphingolipid biosynthesis pathway in age-related muscle homeostasis.

## eLife assessment

This **solid** study presents **valuable** insights into the role of Cers1 on skeletal muscle function during aging, although further substantiation would help to fully establish the experimental assertions. It examines an unexplored aspect of muscle biology that is a relevant opening to future studies in this area of research.

## Introduction

We are currently experiencing the biggest demographic shift in human history with a prospective doubling of the 65+year-old population by the year 2050, thereby constituting almost a quarter of the world's population for the first time (*Bloom et al., 2015*). While the increase in human lifespan

could provide unprecedented opportunities for aged individuals, healthspan is not quite following this trend: One-fifth of an individual's life will be lived with morbidity and reduced mobility (*Partridge et al., 2018*). Health burden escalates with age, which not only incapacitates the elderly but also puts tremendous stress on the healthcare system exemplified by the 20% increased healthcare spending due to the growing elderly population in the last 10 years alone (*Atella et al., 2019*). Hence, understanding the underpinnings of aging is an important first step to inform development of biological tools to promote healthy aging.

Skeletal muscle is one of the organs most affected by aging. Three to 8% muscle mass and strength are lost yearly after 30 years of age and 6–15% after 65 years of age during normal human aging (*Melton, 2000*). Exacerbated age-related loss of muscle mass and strength has recently been recognized as the disease sarcopenia, which predicts and correlates with mortality (*Cruz-Jentoft et al., 2010*). The importance of sarcopenia is further highlighted by the high prevalence of sarcopenia in the common 60+year-old population (10–20%; *Shafiee et al., 2017*), especially considering that sarcopenia is one of the leading causes for frailty, loss of independence and reduced quality of life (*Rizzoli et al., 2013*). Several mechanisms have been implicated in the skeletal muscle aging process, including reduced mitochondrial bioenergetics coupled with increased levels of reactive oxygen species, dysfunctional proteostasis, reduced hormone levels and signaling, nutritional intake, increased, inflammation, loss of motor units, as well as reduced capacities for muscle regeneration and maturation/synthesis (for a review, see *Sartori et al., 2021*). Recent evidence further suggests the involvement of sphingolipids in skeletal muscle homeostasis upon aging (*Tan-Chen et al., 2020*).

Sphingolipids are bioactive lipids exerting pleiotropic cellular functions such as inflammation, proliferation, myelination, cell growth, and cell death (*Hannun and Obeid, 2018*). Ceramides are the building blocks of most complex sphingolipids, which contain a sphingoid base (typically 18 carbon dihydrosphingosine or sphingosine) attached to a variable length fatty acyl side-chain. While sphingolipids have been shown to be involved in skeletal muscle bioenergetics (*Turner et al., 2018*; *Park et al., 2016*), insulin sensitivity (*Holland et al., 2007*) and diabetes (*Zhao et al., 2007*), the contribution of ceramides to skeletal muscle aging is not well understood. We recently reported that the first enzyme of the de novo sphingolipid biosynthesis pathway (SPT) is involved in skeletal muscle aging (*Laurila et al., 2022b*). However, less is known about the involvement of ceramide synthases (Cers) in muscle aging, despite their prominent role in membrane homeostasis and cellular signaling. Six ceramide synthases (Cers1-6) catalyze ceramide synthesis in mammals resulting in a range of ceramides from C14:0 to C36:0 (*Turner et al., 2018*). Depending on the enzyme, different length fatty acyl-coenzyme A (CoA) are transferred to the amine group of the sphingoid base (*Park et al., 2014*). For example, Cers1 only uses 18 carbon (C18) fatty acids to create C18 (d18:1/18:0) ceramides, whereas Cers2 predominantly synthesizes d18:1/24:0 (C24:0) and d18:1/24:1 (C24:1) ceramides and Cers5 uses C16 acyl-CoA substrates for acylation of the free primary amine group of sphingoid bases to form the corresponding C16:0 ceramides (*Mizutani et al., 2005*). *Cers1* and *Cers5* are the two most expressed ceramide synthases in skeletal muscle (*Levy and Futerman, 2010*). Reducing Cers1 function by genetic or pharmacological means has been shown to reduce adiposity and body weight upon exposing mice to a high fat diet (*Turner et al., 2018*; *Gosejacob et al., 2016*). Moreover, BMI was found to correlate with C18:0 ceramide species, pointing towards Cers1 as a desirable target for metabolic interventions (*Weir et al., 2013*). On the other hand, recent evidence demonstrates that skeletal muscle-specific deletion of Cers1 in young mice reduces muscle fiber size and force, suggesting that Cers1 is needed for proper skeletal muscle function in young mice (*Tosetti et al., 2020*). However, Cers1' involvement during organismal aging, and particularly its mechanistic function and relevance in human skeletal muscle is not well characterized.

In the present study, we show that *CERS1* is involved in the age-related loss in muscle mass and function. Performing unbiased gene set enrichment analyses of *CERS1*-correlating transcripts in human skeletal muscle, we discover that *CERS1* levels correlate most strongly with muscle function and myogenesis. Our experiments validate the increase in *CERS1* abundance in mouse and human skeletal muscle cells during myogenic differentiation. Conversely, the selective pharmacological inhibitor of *CERS1*, P053 (*Turner et al., 2018*), impairs myogenic maturation across several timepoints by blunting expression of myogenic regulatory factors and consequently, myosin heavy- and light chains. In aged mice, P053 treatment or AAV9-mediated silencing of *Cers1* in skeletal muscle attenuates C18:0 /C18:1 ceramide and dihydroceramide species, and exacerbates the age-related decline in

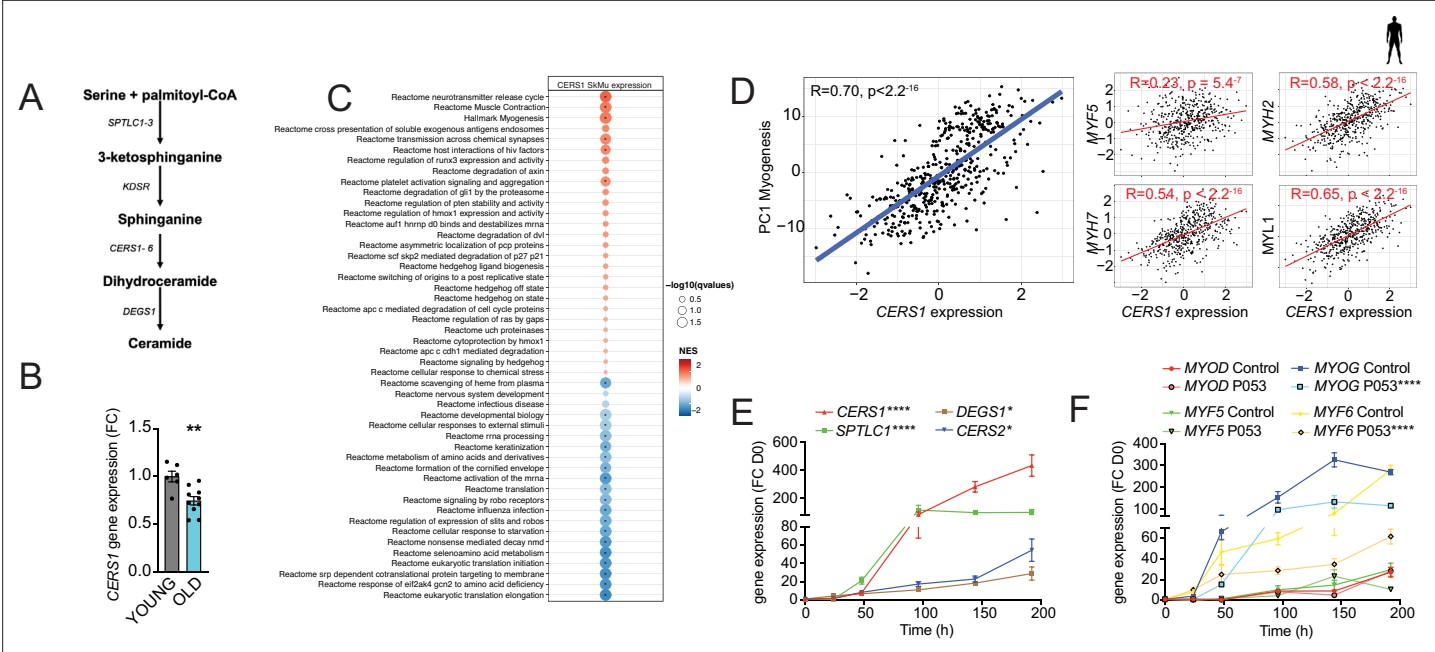

**Figure 1.** CERS1-correlated pathways in human skeletal muscle associate with muscle function and myogenesis. (**A**) Overview over the de novo sphingolipid biosynthesis pathway. (**B**) *CERS1* expression in young (22±3.61 yo) and old (74±8.4 yo) human skeletal muscle cells (n=3–5 biological replicates). (**C**) Gene set enrichment analysis of *CERS1* mRNA correlated transcripts in the human skeletal muscle from the Genotype-Tissue Expression (GTEx) dataset (n=469). Normalized effect size (NES). (**D**) Spearman correlation between skeletal muscle expression of *CERS1* and the first principal component (PC1) of the myogenesis pathway (left) from (**C**) and expression of key genes involved in myogenesis (right). (**E–F**) Gene expression of enzymes involved in the sphingolipid de novo synthesis pathway (**E**) and myogenic regulatory factors (**F**) upon horse serum induced differentiation of primary human skeletal muscle cells (n=2–3 biological replicates). The pharmacological inhibitor P053 or DMSO was used at 1 uM in (**E–F**). Data: mean ± SEM. Unpaired Student's t-test and two-way ANOVA determined significance. *p<0.05, **p<0.01, ****p<0.0001.

The online version of this article includes the following figure supplement(s) for figure 1:

**Figure supplement 1.** *Cers1* regulates muscle contraction in human muscle and myoblast maturation.

muscle fiber size while elevating inflammation and fibrosis. These morphological alterations coincide with reduced myogenesis and reduced muscle mass and function. Lenti- or adenovirus-mediated *CERS1* inhibition specifically in mouse or human muscle cells recapitulates maturation defects of muscle cells. Pharmacological or genetic inhibition of the *CERS1* orthologue *lagr-1* in *Caenorhabditis elegans* similarly impairs age-related muscle function and morphology suggesting conserved CERS1 function across species. Finally, we identify single nucleotide variants in the genomic CERS1 locus, which reduce *CERS1* levels in skeletal muscle and reduce muscle grip strength as well as overall health in humans.

## Results

### *CERS1* expression correlates with muscle contraction and myogenesis

The sphingolipid synthesis pathway produces ceramides and other sphingolipids by using fatty acids and amino acids as substrates (**Figure 1A**). SPT converts L-serine and palmitoyl-CoA to 3-ketosphinganine, which is rapidly converted to sphinganine. Coupling of sphinganine to long-chain fatty acid is accomplished by one of 6 distinct mammalian ceramide synthases. Among these, *Cers1* expression in skeletal muscle tissue was shown to correlate negatively with aging (**Tosetti et al., 2020**). To expand on this finding and assess whether the age-related decline in *CERS1* abundance is mediated specifically by skeletal muscle cells, we measured *CERS1* expression in primary muscle cells isolated from young and old human donors. We found reduced *CERS1* transcript levels in aged compared to young myoblasts which is in line with the previous report in whole muscle tissue (**Tosetti et al., 2020; Figure 1B**).

To study the role of *CERS1* in human skeletal muscle, we performed unbiased gene set enrichment analyses of *CERS1* correlating transcripts in human skeletal muscle biopsies. Molecular pathways involved in muscle contraction and myogenesis were among the strongest correlating pathways (*Figure 1C*). In line with this finding, we also observed significant positive correlations of *CERS1* expression with transcripts involved in myogenesis (*MYF5, MYH2, MYH7, MYL1*; *Figure 1D*) and muscle contraction (*TPM2, ACTA1, TMOD2, TNN1*; *Figure 1—figure supplement 1A*). Assaying the myogenic differentiation process in mouse and human muscle cells confirmed the upregulation of *Cers1/CERS1* at several timepoints once muscle differentiation was induced (*Figure 1—figure supplement 1B* and *Figure 1E*). CERS1 has been shown to synthesize C18:0 and C18:1 ceramides (*Mizutani et al., 2005*; *Ginkel et al., 2012*; *Venkataraman et al., 2002*). Measurement of ceramide levels during myogenic differentiation confirmed particularly the increase of C18:1 during mouse myoblast maturation (*Figure 1—figure supplement 1C*), whereas C14:0, C22:0, as well as C18:0 ceramide species were found to be induced during human myoblast differentiation suggesting some species-specific effects (*Figure 1—figure supplement 1D*).

Importantly, the *CERS1*-specific pharmacological inhibitor P053, which has been shown to reduce fat mass (*Turner et al., 2018*), inhibited mouse and human skeletal muscle cell differentiation as evidenced by downregulated expression of the early expressed myogenic transcription factors *Myog/MYOG* and *Myf6/MYF6* (*Figure 1—figure supplement 1E* and *Figure 1F*) as well as of myosin light- and heavy chains (*Figure 1—figure supplement 1F–G*). In aggregate, these findings suggest that *CERS1* is involved in skeletal muscle cell maturation.

## Pharmacological inhibition of *Cers1* impairs skeletal muscle in aged mice

We next assessed the systemic effect of the pharmacological *Cers1* inhibitor P053 on skeletal muscle homeostasis in aged mice. As expected, systemic administration of P053 over 6 months (*Figure 2A*) reduced C18:0 and C18:1 ceramides in skeletal muscle tissue (*Figure 2B*), suggesting that P053 indeed targeted Cers1 as previously shown (*Turner et al., 2018*). Interestingly, this was coupled with an increase in C24:0 /C24:1 ceramides (*Figure 2B*) and C24:0 /C24:1 dihydroceramides (*Figure 2—figure supplement 1A*), which are synthesized by Cers2 (*Mizutani et al., 2005*; *Ben-David et al., 2011*; *Laviad et al., 2008*). This might reflect a compensatory mechanism to maintain the flow of sphinganine and fatty acid substrates. An increase in C24:0 /C24:1 ceramide and dihydroceramides upon *Cers1* inhibition is in line with a previous reports on *Cers1* inhibition (*Turner et al., 2018*; *Tosetti et al., 2020*). Laminin staining (*Figure 2C*) followed by minimum ferret diameter measurement (*Figure 2—figure supplement 1B*) and fiber size distribution (*Figure 2D*) showed that fiber size was reduced upon aging with a frequency shift towards smaller muscle fibers. Treatment with P053 exacerbated this age-related reduction in muscle fiber size. We next automated the pipeline to detect skeletal muscle fibers in mouse tissue sections by custom training and application of the recently described deep learning-based segmentation algorithm cellpose (*Stringer et al., 2021*). Ground truth data, model training and validation, as well as the script to run on the data are publicly available on Zenodo (DOI: https://zenodo.org/records/7041137). Using our trained model in muscle sections stained with laminin and CD45, we were able to capture most muscle fibers (Average Precision: 0.97±0.02, n=4, *Supplementary file 1*). Results showed an age-related decrease in cross-sectional area, which worsened with P053 treatment (*Figure 2E*, top and *Figure 2—figure supplement 1C*) and is in agreement with our manual curation of fiber diameter (*Figure 2C–D*). We also observed an increase in CD45[+] neighboring muscle fibers with age and P053 treatment (*Figure 2E*, bottom). The effect of age and P053 on muscle inflammation was confirmed by another CD45 staining with DAPI showing an increased proportion of CD45[+] cells upon aging, which was exacerbated by P053 administration (*Figure 2—figure supplement 1D*). Analyses of other morphological markers of skeletal muscle aging, fibrosis and centralized nuclei, revealed that pharmacological inhibition of *Cers1* led to increased fibrosis (*Figure 2F*) and occurrence of centralized nuclei (*Figure 2G*). In line with our findings of differentiating skeletal muscle cell cultures (*Figure 1F* and *Figure 1—figure supplement 1E–G*), P053 treatment downregulated expression of myogenic regulatory factors and of myosin heavy chain transcripts in mice (*Figure 2—figure supplement 1E*). Of note, P053-mediated alterations in skeletal muscle morphology and expression of myogenic factors were correlated with impairments in skeletal muscle mass and function. Lean body mass was reduced upon aging and aged mice treated with P053

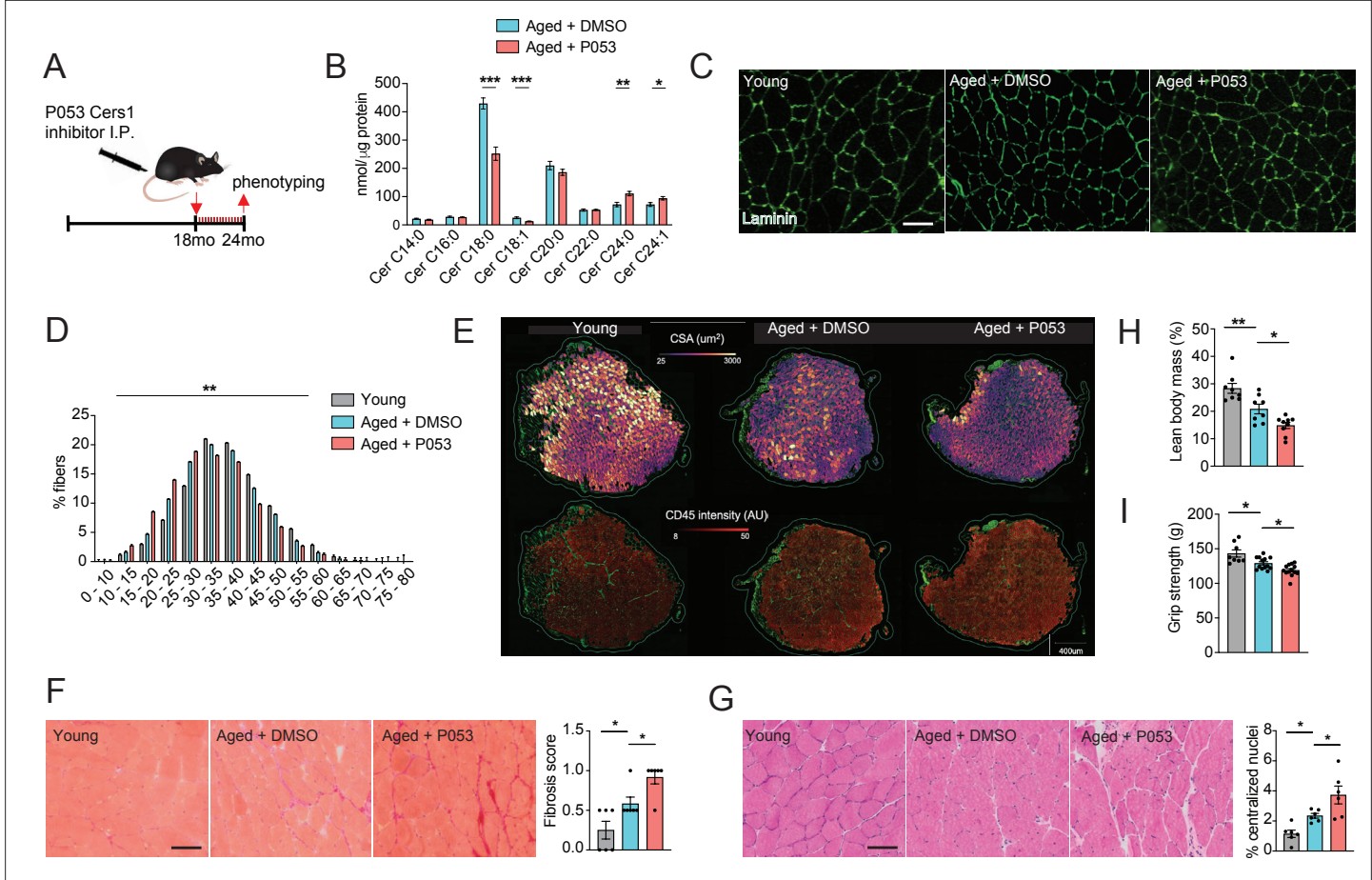

**Figure 2.** P053 administration inhibiting *Cers1* in aged mice deteriorates skeletal muscle function and morphology. (**A**) Overview over the experimental pipeline that was used to administer P053 in aged C57BL/6 J mice using intraperitoneal injections (I.P) three times a week for 6 months. (**B**) Skeletal muscle ceramide levels in aged mice treated with DMSO or P053 (n=7–8 per group). (**C**) Representative images of laminin-stained tibialis anterior of young control or mice injected with DMSO or P053 (n=5–6 per group). Scale bar, 50 µM. (**D**) Quantification of minimal Ferret diameter distribution in skeletal muscle cross-sections from young control or mice injected with DMSO or P053 (n=5–6 per group). (**E**) Automated detection of laminin/CD45+ stained muscle fibers from young control or aged mice injected with DMSO or P053 showing cross-sectional area (top) and inflammation-adjacent signal intensities (bottom). (**F**) Representative brightfield images and quantification of Sirus red stained muscle sections from young and aged mice treated with DMSO or P053 (n=6 per group). (**G**) Representative brightfield images and quantification of muscle cross-sections from young and aged mice treated with DMSO or P053 stained with hematoxylin/eosin (n=6 per group). (**H–I**) Phenotyping measurements of young control and aged mice injected with DMSO or P053 showing (**H**) lean body mass and (**I**) grip strength (n=7–14 per group). Data: mean ± SEM. Unpaired Student's t-test and one-way ANOVA determined significance. *p<0.05, **p<0.01, ***p<0.001.

The online version of this article includes the following figure supplement(s) for figure 2:

**Figure supplement 1.** *Cers1* inhibition exacerbates mouse skeletal muscle aging.

presented further reductions in muscle mass (*Figure 2H*), which is in line with reduced fiber diameters (*Figure 2C–D*) and cross-section area (*Figure 2—figure supplement 1C*). Measurement of functional muscle parameters revealed that P053 administration worsened the effect of aging by reducing grip strength (*Figure 2I*) and treadmill running performance in aged mice (*Figure 2—figure supplement 1F–G*). Our observation of pharmacologic *Cers1* inhibition in aged mice is in line with a previous study where genetic knockout of *Cers1* in young mice impaired muscle contractility (*Tosetti et al., 2020*).

## Muscle- specific genetic inhibition of *CERS1* impairs aged muscle and blunts myogenesis

Systemic administration of P053 in aged mice might have exerted effects on other tissues that affect aged skeletal muscle. Moreover, pharmacological strategies might have undesired off target effects. Therefore, we investigated the effect of direct genetic inhibition of *Cers1* in aged mouse skeletal

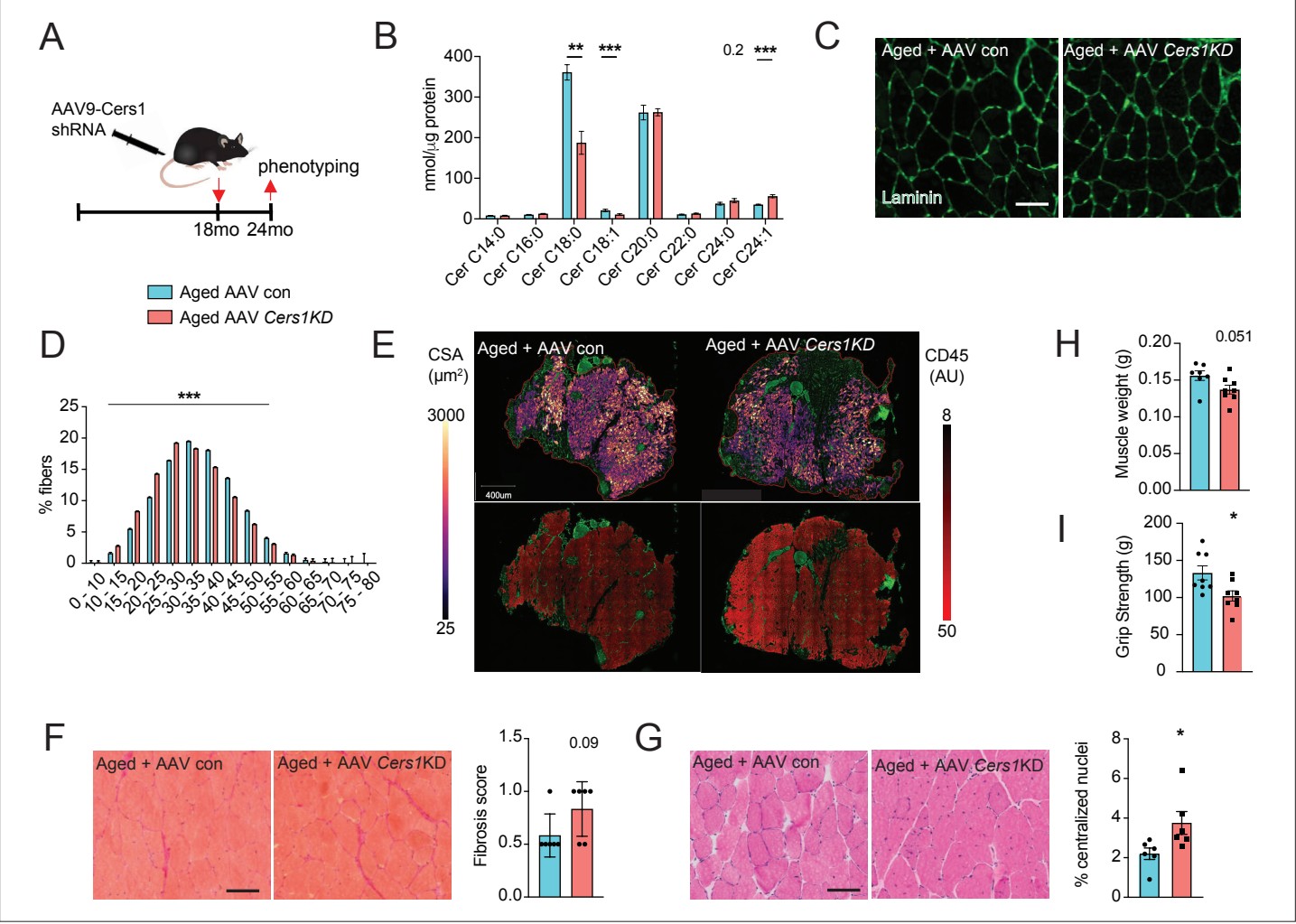

**Figure 3.** Adeno-associated virus 9 (AAV9)-mediated knockdown of *Cers1* expression in aged skeletal muscle reduces skeletal muscle function and morphology. (**A**) Schematic showing a single gastrocnemius intramuscular injection of adeno-associated virus particles containing short hairpin RNA against *Cers1* in aged C57BL/6 J mice. (**B**) Skeletal muscle ceramide levels in aged mice intramuscularly injected with shRNA targeting *Cers1* (n=6–8). (**C**) Representative images of laminin-stained aged mice muscle cross-sections intramuscularly injected with shRNA targeting *Cers1* (n=6 per group). Scale bar, 50 μM. (**D**) Quantification of minimal Ferret diameter distribution in aged mice intramuscularly injected with shRNA targeting *Cers1* (n=6 per group). (**E**) Automated detection of laminin/CD45+ stained skeletal muscle fibers from aged mice intramuscularly injected with shRNA-*Cers1* showing cross-sectional area (top) and inflammation-adjacent signal intensities (bottom). (**F**) Representative brightfield images and quantification of Sirus red stained skeletal muscle cross-sections from mice intramuscularly injected with AAV9 particles containing scramble, or shRNA targeting *Cers1* (n=6 per group). (**G**) Representative brightfield images and quantification of hematoxylin/eosin-stained muscle cross-sections from mice intramuscularly injected with AAV9 particles containing scramble, or shRNA targeting *Cers1* (n=6 per group). (**H–I**) Phenotyping measurements of aged mice intramuscularly injected with shRNA-*Cers1* or a scramble shRNA with (**H**) gastrocnemius muscle weight and (**I**) grip strength (n=6–8 per group). Data: mean ± SEM. Unpaired Student's t-test determined significance. *p<0.05, **p<0.01, ***p<0.001.

The online version of this article includes the following figure supplement(s) for figure 3:

**Figure supplement 1.** *Cers1* silencing deteriorates age-related running, ceramides and blunts myogenesis in myoblasts.

muscle. A single intramuscular injection of AAV9 with short hairpin RNA (shRNA) against mouse *Cers1* (*Figure 3A*) in 18-month-old mice reduced *Cers1* mRNA expression by ≈ 50% at 24 months of age (*Figure 2—figure supplement 1H*) and hence, reduced C18:0 and C18:1 ceramide metabolites (*Figure 3B*). Similarly to our observation in P053-treated animals, genetic inhibition of *Cers1* in aged skeletal muscle tended to increase Cers2-derived very long chain ceramides (*Figure 3B*) and dihydro-ceramides (*Figure 2—figure supplement 1I*). Genetic silencing of *Cers1* reduced average skeletal muscle fiber size (*Figure 3C* and *Figure 2—figure supplement 1J*), shifting muscle fibers towards smaller fibers (*Figure 3D*). This was in line with our custom cellpose model for fiber detection in muscle

cross-sections showing that *Cers1*-deficient, aged muscles had smaller cross-sectional area (*Figure 3E*, top and *Figure 2—figure supplement 1K*), and showed more signs of inflammation (*Figure 3E*, bottom and *Figure 2—figure supplement 1L*). Furthermore, ablation of *Cers1* in aged muscle showed a trend towards increased fibrosis (*Figure 3F*) and more centralized nuclei (*Figure 3G*). We also found that genetic targeting of *Cers1* downregulated expression of myogenic regulatory factors and myosin heavy chains (*Figure 3—figure supplement 1A*). These findings were coupled with reduced muscle mass (*Figure 3H*), as well as reduced muscle grip strength (*Figure 3I*) and running performance in *Cers1*-deficient aged mice (*Figure 3—figure supplement 1B–C*). Reduced grip strength upon *Cers1* inhibition in aged mice is in line with reduced ex vivo contraction capacity of young muscles deficient of *Cers1* (*Tosetti et al., 2020*). Overall, our results indicate that genetic, AAV9-mediated silencing of *Cers1* in skeletal muscle of aged mice deteriorates skeletal muscle morphology and function and therefore, recapitulate the effects observed by pharmacological inhibition of Cers1.

## *Cers1* mediated deterioration of myogenesis is specific to skeletal muscle cells

Several different cell types reside within human and mouse skeletal muscle tissue (*Rubenstein et al., 2020*). To investigate whether the observed detrimental effects of genetic *Cers1* inhibition on muscle cell maturation are specific to skeletal muscle cells, we generated mouse and human skeletal muscle cells deficient of *Cers1/CERS1*. Lentivirus-mediated silencing of *Cers1* using shRNA in mouse C2C12 muscle progenitor cells reduced *Cers1* expression (*Figure 3—figure supplement 1D*) and with it, C18:0 /C18:1 ceramide species (*Figure 3—figure supplement 1E*). This was coupled by an increase in potentially toxic very long chain ceramides (*Figure 3E*) and dihydroceramides (*Figure 3—figure supplement 1F*). Differentiation of these *Cers1*-deficient myoblasts towards myotubes revealed diminished myogenesis evidenced by immunostaining (*Figure 3—figure supplement 1G*), which shows reduced myotube diameter, myotube area and number of multinucleated myotubes (*Figure 3—figure supplement 1H*). Expression profiling revealed reduced transcript expression of myogenic differentiation markers (*Figure 3—figure supplement 1I*), further suggesting that *Cers1* is indispensable for proper myogenic differentiation of mouse muscle cells. Notably, *Cers1* inhibition did not affect protein synthesis but upregulated the Myostatin pathway including the Foxo3/Smad3 transcription factors and its downstream E3 ubiquitin ligases, suggesting an increase in protein degradation (*Figure 4—figure supplement 1A–D*). We next silenced *CERS1* in isolated primary human myoblasts using adenovirus-mediated delivery of a shRNA construct targeting the human *CERS1* transcript (*Figure 4—figure supplement 1E*). As expected, reducing *CERS1* expression in human primary myoblasts reduced C18:0 /C18:1 ceramides (*Figure 4A*). Similar to mouse tissue and cells, *CERS1* inhibition increased very long chain C24:0 /C24:1 ceramides (*Figure 4A*) and C24:0 /C24:1 dihydroceramides (*Figure 4—figure supplement 1F*) and impaired myogenic differentiation, as indicated by immunostaining (*Figure 4B*) of differentiating myoblasts showing smaller myotube diameter, myotube area and reduction in multinucleated myotubes (*Figure 4—figure supplement 1G–I*) and expression profiling (*Figure 4C*). Therefore, *CERS1* appears indispensable for intact myogenic maturation in mouse and human myoblasts.

## Inhibition of the *CERS1* orthologue *lagr-1* in *C. elegans* deteriorates healthspan

Maintaining proper muscle function is vital to healthy aging and potent modulators of this process are typically conserved across species. Hence, the effect of inhibiting Cers1 on skeletal muscle function and morphology upon *C. elegans* aging was evaluated. We used the RW1596 transgenic worm strain expressing GFP under the control of the muscle-specific *myo3p* worm promoter, which allowed us to visualize muscle fibers as shown previously (*Romani et al., 2021*). Exposure of *C. elegans* to P053 mixed within the agar (50 μM and 100 μM) reduced muscle function at the onset of worm aging (day 5) when movement distance and speed are declining (*Figure 4D*, *Figure 5—figure supplement 1A–B*). In line with our observation of reduced muscle function when using P053 in mice, imaging muscle fibers of P053-treated worms showed deteriorated muscle morphology at day 5 as shown by the presence of rigged fibers compared to the smoother muscle fibers observed in the control worms that were exposed to DMSO vehicle (*Figure 4E*). We next evaluated the effect of genetic inhibition of the CERS1 orthologue *lagr-1* on worm muscle function and morphology. Feeding RW1596

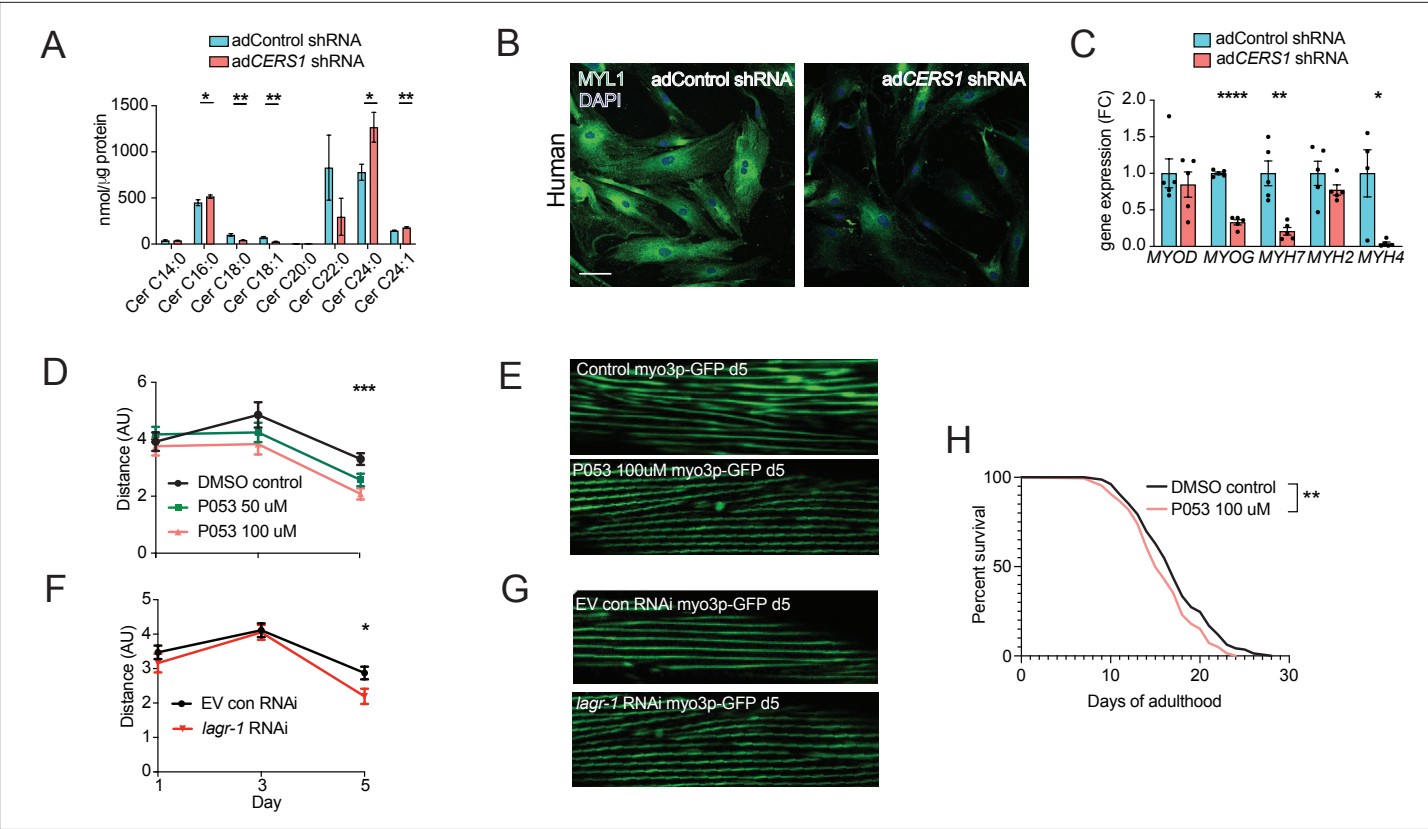

**Figure 4.** Inhibition of *Cers1* in muscle cells impairs myogenic differentiation and C. *elegans* healthspan. (**A**) Primary human skeletal muscle cell ceramide levels infected with adenovirus containing silencing RNA targeting *CERS1* (n=5–6 per group). (**B**) Representative confocal immunocytochemistry images of differentiating human primary muscle cells deficient of *CERS1* (n=4 per group, left). Scale bar, 50 μM. (**C**) Gene expression profiling of differentiating human primary muscle cells deficient of *CERS1* (n=4–5 per group, right). (**D**) Traveled distance in *C. elegans* treated with DMSO, 50uM P053 or 100uM P053 (n=40–57 per group). (**E**) Representative confocal muscle images of RW1596 *C. elegans* (myo3p-GFP) treated with DMSO or 100uM P053 (n=6 per group). (**F**) Traveled distance in *C. elegans* (myo3p-GFP) treated with EV control, or RNAi against *lagr-1* (n=46–56 per group). (**G**) Representative confocal muscle images of RW1596 *C. elegans* (myo3p-GFP) treated with control, or RNAi against *lagr-1* (n=6 per group). (**H**) Lifespan of *C. elegans* treated with control, or 100uM P053 (n=144–147 per group). Data: mean ± SEM. Unpaired Student's t-test, ANOVAs and log-ranked Mantel-Cox determined significance. *p<0.05, **p<0.01, ***p<0.001, ****p<0.0001.

The online version of this article includes the following figure supplement(s) for figure 4:

**Figure supplement 1.** Adenovirus-mediated silencing of *CERS1* reduces ceramides and blunts myogenesis in human muscle cells.

worms with an interference RNA (RNAi) targeting the *lagr-1* transcript reduced its expression by ≈80% (*Figure 5—figure supplement 1C*), and reduced worm muscle function as measured by traveled distance and movement speed (*Figure 4F* and *Figure 5—figure supplement 1D*). Assessment of muscle fiber morphology revealed deteriorated muscle fibers in worms treated with *lagr-1* RNAi at day 5 (*Figure 4G*). Moreover, inhibition of Cers1 using P053 reduced worm lifespan (*Figure 4H*) Taken together, these results suggest that pharmacological or genetic inhibition of the *Cers1* orthologue impairs certain hallmarks of healthy worm aging as suggested by reduced motility and muscle fiber morphology. This is most evident at the onset of worm aging, suggesting that observed effects in lagr-1/Cers1 function might be conserved across species.

## *CERS1* genetic variants reduce muscle *CERS1* expression and muscle function in humans

We next sought to relate our functional mouse and worm findings of *Cers1/CERS1* inhibition impacting on muscle function to humans. To this end, we searched for both common (allele frequency >1%) and rare genetic variants (allele frequency <1%) that alter *CERS1* levels either indirectly via regulatory elements, or directly by mutations in the CERS1 coding region, respectively.

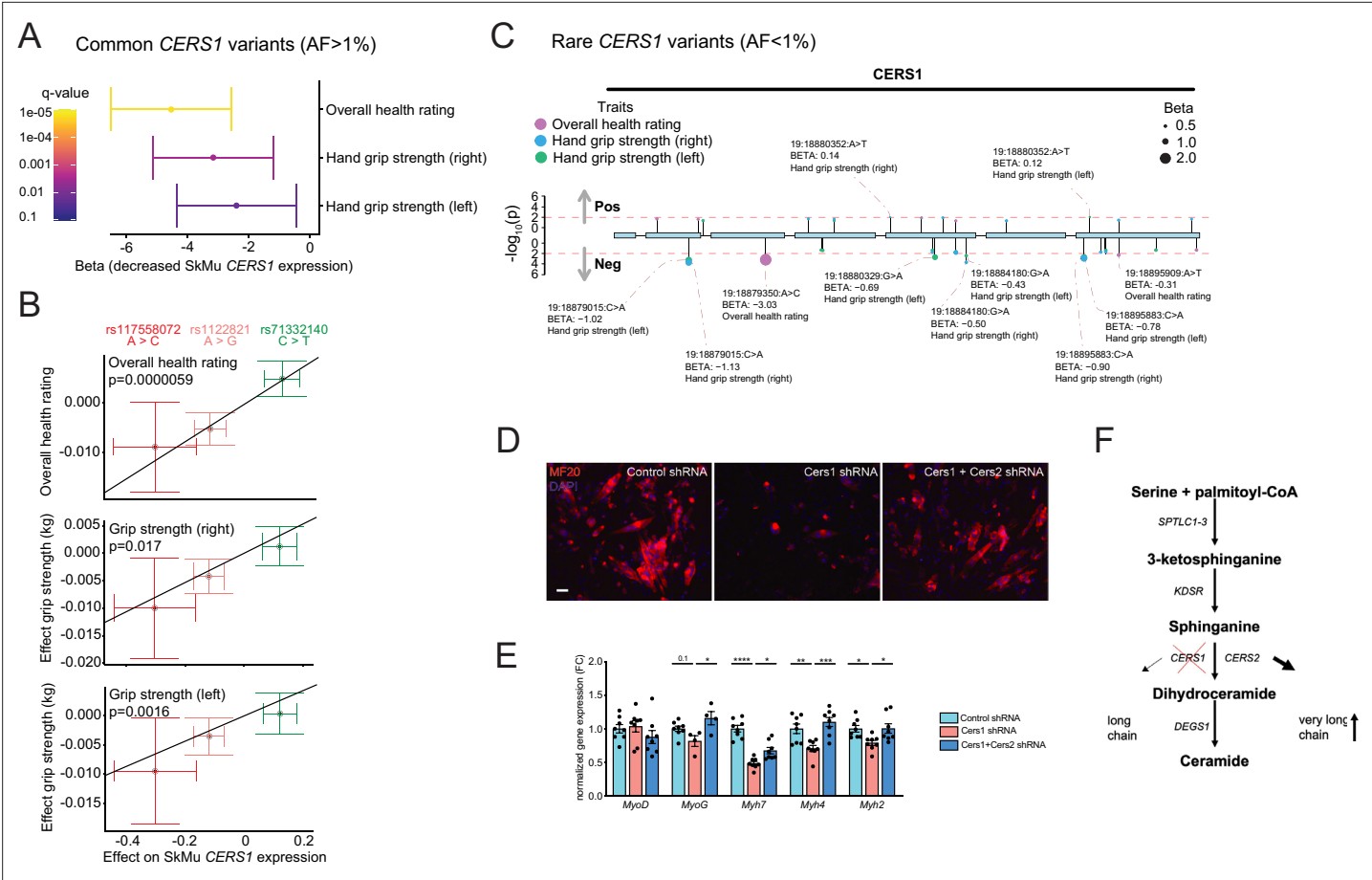

**Figure 5.** Common and rare genetic variants in CERS1 affect muscle function and health in humans. (**A**) Overall result of the Mendelian randomization analysis in the UK biobank cohort using skeletal muscle expression quantitative trait loci (*cis*-eQTL) of *CERS1*. (**B**) Scatter plot showing the effect of the three independent (R² <0.1, see *Figure 5—figure supplement 1E*) *cis*-eQTLs rs117558072, rs1122821, rs71332140 on overall health, right grip strength, and left grip strength in the UK biobank. The slope of the regression line depicts the estimated causal effect with the inverse-variance weighted Mendelian randomization method. (**C**) Lollipop plot depicting rare variants in the coding region of CERS1 and their effects on overall health, right grip strength, and left grip strength in the UK biobank. Colors indicate phenotypes, dot size indicates effect size, red dotted line indicates the suggested cut-off p<0.05. (**D**) Representative immunocytochemistry images of differentiating mouse muscle cells treated with scramble control shRNA, or shRNA targeting *Cers1* alone, or shRNA against *Cers1* and *Cers2*. Scale bar, 50 μM. (**E**) Gene expression profiling of differentiating mouse muscle cells treated with scramble control shRNA, or an shRNA targeting *Cers1* alone, or an shRNA against *Cers1* and *Cers2* (n=4–8 per group). (**F**) Overview over the de novo sphingolipid biosynthesis pathway highlighting the hypothesis that CERS1 inhibition leads to a compensatory upregulation of CERS2, which might inhibit muscle function in aging. Data: mean ± SEM. One-way ANOVA determined significance. *p<0.05, **p<0.01, ***p<0.001.

The online version of this article includes the following figure supplement(s) for figure 5:

**Figure supplement 1.** *CERS1* inhibition is associated with impaired health span.

By exploring the Genotype-Tissue Expression (GTEx) dataset, we discovered three independent (linkage disequilibrium r² <0.1; *Figure 5—figure supplement 1E*), common genetic variants that significantly alter *CERS1* expression in human skeletal muscle (*Figure 5—figure supplement 1F*). The C-allele at rs117558072 and the G-allele at rs1122821 reduce muscle *CERS1* expression, while the T-allele at rs71332140 increases muscle *CERS1* expression. We next evaluated how these skeletal muscle expression quantitative trait loci (eQTLs) might affect overall health, and muscle function by performing Mendelian randomization analysis using the UK biobank traits 'overall health' (data-field 2178), 'right grip strength' (data-field 47) and 'left grip strength' (data-field 46). Combined results of the Mendelian randomization analysis showed that a decrease in muscular *CERS1* expression leads to impaired overall health rating and grip strength (*Figure 5A*). In particular, the *CERS1* muscle expression reducing alleles (rs117558072-C; rs1122821-G) decrease overall health, as well as right and left grip strength. Notably, we find a linear relationship between the eQTL-mediated effect sizes on muscle

*CERS1* expression and the eQTL-mediated effect sizes on the respective phenotypes (*Figure 5B*). Importantly, some of the *CERS1* expression-reducing alleles are common in the human population, as indicated by their allele frequencies ranging from rs117558072-C(2%) to rs1122821-G(44%) and rs71332140-C(77%) and might therefore, contribute to skeletal muscle health in the broad population.

We next evaluated the effect of rare coding variants within the protein-coding exons of CERS1 on the same phenotypes in the UK biobank cohort. Assessing the association of such coding variants with overall health rating, right grip strength and left grip strength, we found that the large majority of the significant (suggestive significant association, p<0.05) variants in the CERS1 coding region (9 out of 11) showed negative effect sizes on overall health rating, right grip strength and left grip strength (*Figure 5C*). Results from these rare variants support the genetic finding of our Mendelian randomization analysis of common variants and therefore, underline the notion that CERS1 might affect overall health and muscle function in humans.

Dysfunctional muscle maturation and increased Cers2 derived C24 ceramides upon Cers1 inhibition, coupled with enhanced myogenic differentiation upon blockage of Cers2-derived C24 ceramides (*Laurila et al., 2022b*) suggest that the observed inhibitory effect of Cers1 knockdown on muscle cell maturation might be mediated by Cers2. Therefore, we tested whether Cers1 knockdown was dependent on Cers2, and knocked down Cers1 and Cers2 simultaneously. Results showed that inhibition of Cers1 together with Cers2 alleviated the impaired differentiation of C2C12 myoblasts that is observed with Cers1 knockdown alone (*Figure 5D–E*, *Figure 5—figure supplement 1G–I*). These findings suggest that very long chain ceramides might drive at least some of the negative muscular consequences observed upon reduced Cers1 function.

In conclusion, our study reveals the necessity of intact *CERS1* expression for muscle cells to undergo muscle maturation, and that disruption of *CERS1* expression might contribute to muscle dysfunction across different species. Due to our consistent observation of increased very long chain C24:0 /C24:1 ceramides and dihydroceramides upon *Cers1/CERS1* inhibition, we suggest a molecular network in which the compensatory upregulation of CERS2 derived very long chain ceramide and dihydroceramide species in aging might accelerate age-related muscle wasting and dysfunction, hence contributing to sarcopenia (*Figure 5F*).

## Discussion

In the present study, we sought to investigate the contribution of CERS1 to the skeletal muscle aging process. Our main findings show that (I) *CERS1* expression correlates positively with skeletal myogenesis and function in human muscle biopsies; (II) intact CERS1 is necessary for proper muscle cell maturation; (III) Cers1 is indispensable for intact organismal muscle aging, shown by deterioration of muscle mass, function and morphology upon genetic/pharmacologic inhibition of Cers1; and (IV) common (5–70% allele frequency) and rare variants in humans affect muscle CERS1 and hence, muscle function and perception of one's own health.

Our findings of reduced fiber size and muscle weight together with reduced muscle strength upon *Cers1* knockdown in aged mice are in line with previous findings by *Tosetti et al., 2020* who show that *Cers1* knockdown in young mice reduced muscle fiber size and contractility. Ceramide levels, and particularly *Cers1*-derived C18-ceramides were previously found elevated in skeletal muscle upon high-fat diet (*Lanza et al., 2013*), streptozotocin-induced diabetes (*Zabielski et al., 2014*) and in obese, diabetic patients (*Bergman et al., 2016*). *Cers1* would therefore appear as an attractive target to improve metabolic outcomes due to its effect on C18 ceramide species. Indeed, the isoform-specific *Cers1* inhibitor P053 reduced C18 ceramide species (while increasing C24 ceramides) exclusively in skeletal muscle, which reduced fat mass in mice fed a high-fat diet (*Turner et al., 2018*). However, our study shows that the improvements in fat mass loss might come at the expense of muscle function/ myogenesis, particularly upon aging. We show that inhibition of *Cers1* reduces myogenesis and exacerbates age-related muscle inflammation and fibrosis – key parameters associated with muscle aging. Furthermore, intact Cers1 might not only be important for skeletal muscle homeostasis in aging; *Cers1* deficiency from birth was previously also reported to cause defects in foliation, progressive shrinkage, and neuronal apoptosis in the cerebellum, which translated to functional deficits including impaired exploration of novel objects, locomotion, and motor coordination (*Ginkel et al., 2012*).

An interesting observation was that *Cers1* inhibition upregulated the levels of C24:0 /C24:1 ceramides in all our *Cers1* targeting strategies and muscle models. This is consistent with previous literature

(**Turner et al., 2018**; **Tosetti et al., 2020**). It suggests a possible compensatory upregulation of *Cers2* to make up for *Cers1* deficiency, or a re-routing of increased sphingoid base substrates towards very long chain (dh)ceramides by Cers2 in the wake of reduced Cers1 activity. Such a re-routing might reflect a cellular adaptation where a tradeoff between cell apoptosis mediated by sphingosine/sphinganine (**Maceyka et al., 2002**; **Ahn and Schroeder, 2010**; **Noack et al., 2014**; **Phillips et al., 2007**) and reduced muscle maturation mediated by very long chain (dh)C24 ceramides, is tilted towards the latter. Of note, compensation effects in brain and liver have also been reported in *Cers2* knockout mice (**Pewzner-Jung et al., 2010**; **Imgrund et al., 2009**), and upregulation of *Cers2* to compensate for *Cers1* deficiency in neurons reduced long chain C24:0 /C24:1 ceramides and hence, ameliorated *Cers1* deficiency-mediated neurodegeneration (**Spassieva et al., 2016**). These studies combined with the dependency of the effects of Cers1 inhibition on Cers2 to reduce myogenic differentiation, therefore suggest that the long chain ceramides C24:0 /C24:1 might exert detrimental effects at least in brain and skeletal muscle tissues. Confirmation of this hypothesis would open new avenues for targeted drug development, as *Cers1* targeting shows conflicting effects with regards to metabolic and muscle health.

We would like to point out that the pharmacologic inhibition of Cers1 was superior to the genetic inhibition, as judged by the reduction in C18 ceramide levels. At the same time, mice treated with the pharmacologic inhibitor displayed more pronounced phenotypes. While it is intriguing to assume that changes in muscle phenotypes are correlated to the degree of ceramide inhibition, we cannot completely rule out multi-organ cross talk(s) of other organs that were affected by the systemic administration of the pharmacological inhibitor.

In summary, our study links *Cers1* with muscle cell maturation and muscle function in the context of aging. Disrupting *Cers1*, and hence the myogenic potential of aged skeletal muscle, exacerbates the age-related decline in skeletal muscle mass, function, and morphology. Given the highly beneficial effects of Sptlc1 inactivation on healthy aging (**Laurila et al., 2022b**), our current study further suggests that a significant part of the benefits of blocking the de novo ceramide biosynthesis pathway might come from inhibiting very long chain ceramides.

## Methods
### In vivo experiments
#### Animal experiments
Young (3 months) and aged (18-month-old) male C57BL/6JRj mice were obtained from Janvier Labs. Mice were randomized to the respective treatment groups according to their body weight. Animals were fed a standard chow diet. All animals were housed in micro-isolator cages in a room illuminated from 7:00AM to 7:00PM with ad libitum access to diet and water. The dose of P053 or DMSO was 5 mg/kg three times a week. Adeno-associated virus (AAV) 9 was bilaterally injected in gastrocnemius muscle at 18 month of age with $2\times10^{11}$ viral particles (Vector Biolabs, United States) containing scramble short hairpin RNA (shRNA) or *Cers1*-targeting shRNA. Five to 6 months after pharmacological or gene-modifying injections, phenotypic tests were performed on the animals. At sacrifice, muscles were removed for histochemical analyses or snap frozen in liquid $N_2$ for biochemical assays. Use of animals for all experimental studies were approved by animal licenses 2890.1 and 3341 in Canton of Vaud, Switzerland and were in compliance with the 1964 Declaration of Helsinki and its later amendments.

#### Endurance running test
After familiarizing with the treadmill, the exercise regimen started at a speed of 15 cm/s. Every 12 min, the speed was increased by 3 cm/s. Mice were considered to have reached their peak exercise capacity, and removed from the treadmill if they received 7 or more shocks (0.2 mA) per minute for 2 consecutive minutes. The distance traveled and time before exhaustion were registered as maximal running distance and time as shown previously (**Laurila et al., 2022a**).

#### Grip strength
Muscle strength was assessed by the grip strength test as previously described (**Wohlwend et al., 2021**). Grip strength of each mouse was measured on a pulldown grid assembly connected to a grip

strength meter (Columbus Instruments). The mouse was drawn along a straight line parallel to the grip, providing peak force. The experiment was repeated three times, and the highest value was included in the analysis.

## Measurement of sphingolipids

Cell pellets (~1.0e6 cells) were extracted by the addition of 100 µL of MeOH spiked with the stable isotope-labeled internal standards (Spa(d17:0), Cer(d18:1/16:0)-d9, Cer(d18:1/18:0)-d7, Cer(d18:1/24:0)-d7 and Cer(d18:1/24:1)-d7). Sample homogenization was performed in the Cryolys Precellys Tissue Homogenizer (2x20 s at 10,000 rpm, Bertin Technologies, Rockville, MD, US) with ceramic beads. The bead beater was air-cooled down at a flow rate of 110 L/min at 6 bar. Homogenized extracts were centrifuged for 15 min at 4000 × $g$ at 4 °C and the resulting supernatants were collected for the LC-MS/MS analysis. Sphingolipids were quantified by LC-MS/MS analysis in positive ionization mode using a 6495 triple quadrupole system (QqQ) interfaced with 1290 UHPLC system (Agilent Technologies), adapted from *Checa et al., 2015*. Briefly, the chromatographic separation was carried out in a Zorbax Eclipse plus C8 column (1.8 µm, 100 mm × 2.1 mm I.D Agilent technologies). Mobile phase was composed of A=5 mM ammonium formate and 0.2% formic acid in water and B=5 mM ammonium formate and 0.2% formic acid in MeOH at a flow rate of 400 µL/min. Column temperature was 40 °C and sample injection volume 2 µL. The linear gradient elution starting from 80% to 100% of B (in 8 min) was applied and held until 14 min. The column was then equilibrated to initial conditions. ESI source conditions were set as follows: dry gas temperature 230 °C, nebulizer 35 psi and flow 14 L/min, sheath gas temperature 400 °C and flow 12 L/min, nozzle voltage 500 V, and capillary voltage 4000 V. Dynamic Multiple Reaction Monitoring (dMRM) was used as acquisition mode with a total cycle time of 500ms. Optimized collision energies for each metabolite were applied. Raw LC-MS/MS data was processed using the Agilent Quantitative analysis software (version B.07.00, MassHunter Agilent technologies). For absolute quantification, calibration curves and the stable isotope-labeled internal standards (IS) were used to determine the response factor. Linearity of the standard curves was evaluated for each metabolite using a 12-point range, in addition, peak area integration was manually curated and corrected when necessary. Sphingolipid concentrations were reported to protein concentrations measured in protein pellets (with BCA, Thermo Fisher Scientific) following the metabolite extraction.

## Histology

Skeletal muscles were harvested from anesthetized mice, and immediately embedded in Thermo Scientific Shandon Cryomatrix and frozen in isopentane, cooled in liquid nitrogen, for 1 min before being transferred to dry ice and stored at –80 °C. Eight µm cryosections were incubated in 4% PFA for 15 min, washed three times for 10 min with PBS, counterstained with DAPI, laminin (1:200, Sigma), CD45 (1:200, Life Technologies), coupled with Alexa-488 or Alexa-568 fluorochromes (Life Technology) and mounted with Dako Mounting Medium. Images was performed with VS120-S6-W slides scanner (Olympus) acquired using a 20 x/0.75 air UPLS APO objective and an Olympus XM10 CCD camera. Three channel images (DAPI, Laminin, and CD45-A568) were acquired defining a focus map based on the Laminin signal for fluorescence imaging whereas hematoxylin and eosin and Sirius Red were imaged with brighfield. Resulting VSI images could be opened in Fiji/ImageJ and QuPath using the BioFormats library. Minimum Feret diameter, and cross-sectional area were determined using the ImageJ software. For fiber size distribution, a minimum of 2000 fibers were used for each condition and measurement. The minimum Feret diameter is defined as the minimum distance between two parallel tangents at opposing borders of the muscle fiber. This measure has been found to be resistant to deviations away from the optimal cross-sectioning profile during the sectioning process.

## Deep learning-based muscle segmentation and quantification

A subset of four muscle cross-section images was manually annotated in multiple regions to serve as ground-truth for cellpose training. In total, 15 areas were used for training and 4 for validation. Annotation and training was done through QuPath using the QuPath cellpose extension (https://github.com/BIOP/qupath-extension-cellpose). Original images were exported as 'raw image plus labeled image' pairs and downsampled 4 x in X and Y (Final pixel size for training: 1.2876 µm/px). The cellpose model was trained with the starting weights of cellpose's 'cyto2' model, for 500 epochs.

Predicted validation images were further run through a validation notebook derived from ZeroCost-DL4Mic (*von Chamier et al., 2021*) in order to assess the quality of the predictions (see *Supplementary file 1*). Segmentation of muscle fibers was performed in two steps. First, the whole cross-section was detected though a gaussian-blurred version of the Laminin signal (sigma = 5 μm), followed by an absolute threshold (threshold value: 5). The resulting region was then fed into the QuPath cellpose extension along with our custom model for individual fiber segmentation. Training images, training and validation scripts and Jupyter notebooks, as well as an example QuPath project are available on Zenodo (DOI: 10.5281/zenodo.7041137).

### *C. elegans* motility assays

*C. elegans* N2 strains were cultured at 20 °C on nematode growth medium agar plates seeded with *E. coli* strain OP50 unless stated otherwise. Synchronous animal populations were generated through hypochlorite treatment of gravid adults to obtain synchronized embryos. These embryos were allowed to develop into adulthood under appropriate conditions. The *Cers1* orthologue *lagr-1* was found on wormbase (https://wormbase.org/). For RNAi experiments, worms were exposed to *lagr-1* RNAi or an empty vector control plasmid using maternal treatment to ensure robust knock down. P053 was dissolved in DMSO, and used at a final concentration of 50 μM and 100 μM. Worms were exposed to P053 starting from L4 stage. P053 or DMSO was added to agar before preparing plates. To ensure permanent exposure to the compound and avoid bacterial contamination, plates were changed twice a week. *C. elegans* movement analysis was performed using the Movement Tracker software as done previously (*Wohlwend et al., 2021*). Measurement of body thrashes was performed as previously described (*Nawa and Matsuoka, 2012*). Briefly, animals grown on plates containing either DMSO or P053 were placed in a 50 μl drop of M9 and the thrashes performed within a 90 s interval were counted.

### *C. elegans* muscle morphology

The transgenic RW1596 strain (*Caenorhabditis* Genetics Center, University of Minnesota) was used for muscle morphology. RW1596 expresses GFP under the control of the worm skeletal muscle promoter myo3 (myo3p-GFP). To perform muscle stainings, ~50 worms were washed in M9, immobilized with 7.5 mM tetramisole hydrochloride (Sigma-Aldrich) and immediately imaged afterwards. Confocal images were acquired with Zeiss LSM 700 Upright confocal microscope (Carl Zeiss AG) under non-saturating exposure conditions. Image processing was performed with Fiji.

### *C. elegans* lifespan

Twenty to 25 worms were placed on nematode growth medium agar plates containing 100μM P053 or DMSO as a control, along with 10μM 5-FU, and seeded with HT115(DE3) bacteria. A minimum of 140 animals were tested per condition. Animals were assessed daily for touch-provoked movement. Worms that died due to internally hatched eggs, an extruded gonad, or desiccation resulting from crawling on the edge of the plates were excluded. Survival assays were repeated twice, and curves generated using the product-limit method of Kaplan and Meier. The log-rank (Mantel–Cox) test was used to evaluate differences between survivals and determine significance.

## In vitro mouse and human studies

### Cell culture and cell transfection

The C2C12 mouse myoblast cell line was obtained from the American Type Culture Collection (CRL-1772TM). C2C12 cells or clones were cultured in growth medium consisting of Dulbecco's modified Eagle's medium (Gibco, 41966–029), 20% Fetal Bovine Serum (Gibco, 10270–106) and 100 U/mL penicillin and 100 mg/mL streptomycin (Gibco, 15140–122). To induce differentiation, FBS was substituted with 2% horse serum (Gibco, 16050–122). Trypsin-EDTA 0.05% (GIBCO, 25300–062) was used to detach cells. Primary human skeletal muscle cells were obtained from Lonza (SkMC, #CC-2561) and the Hospices Civils de Lyon and were cultured in growth medium consisting of DMEM/F12 (Gibco, 10565018), 20% Fetal Bovine Serum (Gibco, 10270–106) and 100 U/mL penicillin and 100 mg/mL streptomycin (Gibco, 15140–122). To induce differentiation, FBS was exchanged with 2% horse serum (Euroclone) and kept in culture. All cells were maintained at 37 °C with 5% $CO_2$. Cell transfections

were done using TransIT-X2 (Mirus) according to the manufacturer's protocol with a 3:1 ratio of transfection agent to DNA. C2C12 cells were grown confluent, and 1 μM P053 or DMSO was added, and cells were kept in growth medium for another 3 days to measure muscle cell differentiation. Protein synthesis was measured according to the instructions of the Click-iT HPG kit (C10428, Thermo Fisher Scientific). All cell lines were regularly tested for mycoplasma and inspected for fungi contamination.

## RNA isolation and real-time qPCR

Tissues were homogenized with Trizol, whereas cells were homogenized, and RNA isolated using the RNeasy Mini kit (QIAGEN, 74106). Reverse transcription was performed with the High-Capacity RNA-to-cDNA Kit (4387406 Thermo Fisher Scientific) as shown previously (*Moreira et al., 2019*). Gene expression was measured by quantitative reverse transcription PCR (qPCR) using LightCycler 480 SYBR Green I Master (50-720-3180 Roche). Quantitative polymerase chain reaction (PCR) results were calculated relative to the mean of the housekeeping gene *Gapdh*. The average of two technical replicates was used for each biological data point. Primer sets for qPCR analyses are shown in *Supplementary file 2*.

## Lentivirus production and transduction

Lentiviruses were produced by cotransfecting HEK293T cells with lenti plasmids expressing scramble shRNA or *Cers1* targeting shRNA (see plasmid list in *Supplementary file 3*), the packaging plasmid psPAX2 (addgene #12260) and the envelope plasmid pMD2G (addgene #12259), in a ratio of 4:3:1, respectively. Transfection medium was removed 24 hr after transfection and fresh medium was added to the plate. Virus containing medium was collected at 48 hr and concentrated using Lenti-X Concentrator (Takara). C2C12 mouse myoblasts were seeded 24 hr prior to infection and then transduced with virus-containing supernatant supplemented with 8 μg/mL polybrene (Millipore) as shown previously (*Wohlwend et al., 2021*). For the delivery of a shRNA construct targeting *CERS1* in human primary myoblasts, adenovirus containing Ad-U6-h-CERS1-shRNA or Ad-U6-scrmb-shRNA (Vector Biolabs) was infected using 1 μL adenovirus per mL cell culture medium at a titer of $5 \times 10^{10}$ PFU/ml.

## Immunocytochemistry

C2C12 cells or Lonza SkMU cells cultured on a sterilized coverslip in six-well plates (Greiner bio-one, CELLSTAR, 657160) were fixed in Fixx solution (Thermo Scientific, 9990244) for 15 min and permeabilized in 0.1% Triton X-100 (Amresco, 0694) solution for 15 min at 20 °C. Cells were blocked in 3% BSA for 1 hr at 20 °C to avoid unspecific antibody binding and then incubated with primary antibody over night at 4 °C with gentle shaking. MyHC was stained using the MF20 primary antibody (1:200, Invitrogen, 14-6503-82) for C2C12 cells and in Lonza muscle cells with a MYL1 antibody (1:140, Thermofisher, PA5-29635). Antibodies used in this study are shown in *Supplementary file 4*. The next day cells were incubated with secondary antibody (Thermo Fisher #A10037 for MF20 and #A-21206 for MYL1) for 1 hr at 20 °C and nuclei were labeled with DAPI. The immunofluorescence images were acquired using either fluorescence or confocal microscopy. The myofusion index was calculated as the ratio of nuclei within myotubes to total nuclei. Myotube diameter was measured for 8 myotubes per image using ImageJ. Myotube area was calculated as the total area covered by myotubes.

## Human studies

### Skeletal muscle gene expression in Genotype-Tissue Expression (GTEx) project

The Genotype-Tissue Expression (GTEx) project version 8 was accessed through accession number dbGaP phs000424.v8.p2 (approved request #10143-AgingX). Transcript expressions and covariates from male human muscle of 469 individuals were extracted from the GTEx Portal (https://gtexportal.org). To remove the known and hidden factors, the Probabilistic Estimation of Expression Residuals (PEER) was applied the expression residuals were used to further analysis. For gene set enrichment analysis, genes were ranked by Spearman correlation coefficients with the gene expression level of *CERS1* based on the Spearman correlation approach, and the *CERS1* enriched gene sets were calculated by GSEA using clusterProfiler R package (version 3.10.1). Gene ontology (GO), Kyoto

Encyclopedia of Genes and Genomes (KEGG), Reactome and Hallmark gene sets were retrieved from the Molecular Signatures Database (MSigDB) using the msigdbr R package (version 7.2.1).

## Mendelian randomization

We used inverse variance-weighted Mendelian randomization to assess the causal effect of *CERS1* expression on muscle-related traits in humans. As exposures, we used the GTEx version 8 cis-eQTLs that were mapped by the GTEx consortium based on the normalized gene expressions in skeletal muscle from 706 deceased human individuals. These summary statistics are publicly available and can be accessed on the internet at the online GTEx webpage portal through https://storage.googleapis.com/adult-gtex/bulk-qtl/v8/single-tissue-cis-qtl/GTEx_Analysis_v8_eQTL.tar. As outcomes, we used 10648 publicly available summary statistics obtained from various sources through the "available_outcomes" function from the TwoSampleMR R package. Outcome IDs can be queried from the ieu gwas database at https://gwas.mrcieu.ac.uk/. We restricted the analysis to independent instrumental variables (genetic variants). We therefore followed an iterative pruning workflow whereby we included the variant with the most significant effect on *CERS1* expression, followed by pruning with a 0.05 squared correlation coefficient ($r^2$) threshold, then again selecting the most significant variant, and so on until no variants are left. This procedure retained three genetic variants: rs117558072 (A>C), rs1122821 (A>G), and rs71332140 (C>T). The causal effect of altered *CERS1* expression was inferred on the 10648 outcomes through inverse-variance weighted Mendelian randomization. The false discovery rate was controlled at the 5% significance level with the Benjamini-Hochberg procedure.

## UK Biobank (UKBB) rare variant analysis

We performed genetic association analyses for hand grip strength and overall health in UK Biobank participants from 379,530 unrelated individuals. To run GWAS on the UK Biobank 450 k Whole Exome Sequencing release (data field 23150) we filtered variants based on allele frequency and Hardy-Weinberg equilibrium. We then used Regenie v3.1.1 and accounted for sex, age, and body mass index. First, population structure estimation was performed using array sequencing to equally represent the whole genome. Second, GWAS burden tests were employed to measure SNP effects on the UKBB phenotypes overall health, right grip strength and left grip strength using the UKBB whole exon sequencing data set. The UKBB data was accessed under the application #48020.

## Statistical analyses

For experimental conditions in which there were two independent factors and multiple comparisons, a factorial ANOVA with subsequent post hoc analysis was performed. Where appropriate, one-way ANOVA with post hoc analysis or t-tests was performed. All statistical analyses were performed using GraphPad Prism (9.4.0, San Diego, CA, USA). Results are reported as means ± standard error of the mean. Statistical significance was set to $p < 0.05$.

# Acknowledgements

We wish to thank the staff of EPFL histology, bioimaging and optics (BIOP), flow cytometry, and animal facilities for technical assistance. We also wish to acknowledge Tony Teav (from Metabolomics Platform at UNIL) for his help with the sphingolipid measurements. This research has been conducted using the UK Biobank Resource. The work in the JA laboratory was supported by grants from the Ecole Polytechnique Federale de Lausanne (EPFL), the European Research Council (ERC-AdG-787702), the Swiss National Science Foundation (SNSF 31003 A_179435), the Fondation Suisse de Recherche sur les Maladies Musculaires (FSRMM) and the Fondation Marcel Levaillant (190917). MW's position was supported by Central Norway Regional Health Authority. BC was supported by FAPESP (2019/22512-0). LJEG was supported by the European Union's Horizon 2020 research and innovation program through the Marie Skłodowska-Curie Individual Fellowship "AmyloAge" (grant agreement no. 896042).

# Additional information

## Competing interests

Pirkka-Pekka Laurila, Johan Auwerx: Inventor on a patent application (WO 2021/058497 A1) related to inhibition sphingolipid synthesis for the treatment of age-related diseases. The other authors declare that no competing interests exist.

## Funding

| Funder | Grant reference number | Author |
|---|---|---|
| European Research Council | ERC-AdG-787702 | Ludger JE Goeminne<br>Tanes Lima<br>Ioanna Daskalaki<br>Xiaoxu Li<br>Giacomo von Alvensleben |
| Swiss National Science Foundation | 31003A_179435 | Martin Wohlwend<br>Pirkka-Pekka Laurila<br>Ludger JE Goeminne<br>Tanes Lima<br>Ioanna Daskalaki<br>Xiaoxu Li<br>Giacomo von Alvensleben |
| Fondation Suisse de Recherche sur les Maladies Musculaires | | Martin Wohlwend<br>Pirkka-Pekka Laurila<br>Ludger JE Goeminne<br>Tanes Lima<br>Ioanna Daskalaki<br>Xiaoxu Li<br>Giacomo von Alvensleben |
| Fondation Marcel Levaillant | 190917 | Martin Wohlwend<br>Pirkka-Pekka Laurila<br>Ludger JE Goeminne<br>Tanes Lima<br>Ioanna Daskalaki<br>Xiaoxu Li<br>Giacomo von Alvensleben |
| Fundação de Amparo à Pesquisa do Estado de São Paulo | 2019/22512-0 | Barbara Crisol |
| Central Norway Regional Health Authority | | Martin Wohlwend |
| Horizon 2020 - Research and Innovation Framework Programme | Marie Skłodowska-Curie Individual Fellowship "AmyloAge" 896042 | Ludger JE Goeminne |

The funders had no role in study design, data collection and interpretation, or the decision to submit the work for publication.

## Author contributions

Martin Wohlwend, Conceptualization, Resources, Data curation, Formal analysis, Supervision, Funding acquisition, Investigation, Visualization, Methodology, Writing – original draft, Project administration, Writing – review and editing; Pirkka-Pekka Laurila, Conceptualization, Data curation, Formal analysis; Ludger JE Goeminne, Data curation, Formal analysis, Validation; Tanes Lima, Giacomo von Alvensleben, Amélia Lalou, Stephen Butler, Data curation; Ioanna Daskalaki, Barbara Crisol, Renata Mangione, Data curation, Formal analysis, Visualization; Xiaoxu Li, Data curation, Formal analysis, Investigation; Hector Gallart-Ayala, Julijana Ivanisevic, Data curation, Methodology; Olivier Burri, Software, Visualization; Jonathan Morris, Resources; Nigel Turner, Resources, Data curation; Johan Auwerx, Conceptualization, Resources, Supervision, Funding acquisition, Writing – original draft, Writing – review and editing

## Author ORCIDs
Martin Wohlwend (iD) http://orcid.org/0000-0001-9851-0364
Xiaoxu Li (iD) http://orcid.org/0000-0001-5121-9190
Giacomo von Alvensleben (iD) http://orcid.org/0000-0001-7231-0276
Hector Gallart-Ayala (iD) http://orcid.org/0000-0003-2333-0646
Jonathan Morris (iD) https://orcid.org/0000-0002-5109-9069
Johan Auwerx (iD) https://orcid.org/0000-0002-5065-5393

## Ethics
Use of animals for all experimental studies were approved by animal licenses 2890.1 and 3341 in Canton of Vaud, Switzerland and were in compliance with the 1964 Declaration of Helsinki and its later amendments.

Reviewer #1 (Public review): https://doi.org/10.7554/eLife.90522.3.sa1
Reviewer #2 (Public review): https://doi.org/10.7554/eLife.90522.3.sa2
Author response https://doi.org/10.7554/eLife.90522.3.sa3

---

# Additional files

## Supplementary files
- Supplementary file 1. Cellpose training quality control.
- Supplementary file 2. List of mouse and human qPCR primers.
- Supplementary file 3. List of plasmids.
- Supplementary file 4. List of antibodies.
- MDAR checklist
- Source data 1. Numerical source data for *Figures 1–5* and accompanying figure supplements.

## Data availability
All data generated or analysed during this study are included in the manuscript and supporting files. The muscle fiber segmentation model is available on https://zenodo.org/records/7041137.

The following dataset was generated:

| Author(s) | Year | Dataset title | Dataset URL | Database and Identifier |
|---|---|---|---|---|
| Wohlwend M, Burri O, Auwerx J | 2024 | Cellpose training data and scripts from Inhibition of CERS1 in aging skeletal muscle exacerbates age-related muscle impairments | https://doi.org/10.5281/zenodo.7041136 | Zenodo, 10.5281/zenodo.7041136 |

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
