## [Editor Report · eLife assessment]

This **solid** study presents **valuable** insights into the role of Cers1 on skeletal muscle function during aging, although further substantiation would help to fully establish the experimental assertions. It examines an unexplored aspect of muscle biology that is a relevant opening to future studies in this area of research.

---

## [Referee Report · Reviewer #1 (Public review)]

Summary:

The authors identified that genetically and pharmacological inhibition of CERS1, an enzyme implicated in ceramides biosynthesis worsen muscle fibrosis and inflammation during aging.

Strengths:

The study points out an interesting issue on excluding CERS1 inhibition as a therapeutic strategy for sarcopenia. Overall, the article it's well written and clear.

---

## [Referee Report · Reviewer #2 (Public review)]

Summary:

The manuscript by Wohlwend et al. investigates the implications of inhibiting ceramide synthase Cers1 on skeletal muscle function during aging. The authors propose a role for Cers1 in muscle myogenesis and aging sarcopenia. Both pharmacological and AAV-driven genetic inhibition of Cers1 in 18-month-old mice lead to reduced C18 ceramides in skeletal muscle, exacerbating age-dependent features such as muscle atrophy, fibrosis, and center-nucleated fibers. Similarly, inhibition of the Cers1 orthologue in *C. elegans* reduces motility and causes alterations in muscle morphology.

Strengths:

The study is well-designed, carefully executed, and provides highly informative and novel findings that are relevant to the field.

---

## [Author Response]

**Reviewer #1 (Public Review):**
Summary:The authors identified that genetically and pharmacological inhibition of CERS1, an enzyme implicated in ceramides biosynthesis worsen muscle fibrosis and inflammation during aging.Strengths:The study points out an interesting issue on excluding CERS1 inhibition as a therapeutic strategy for sarcopenia. Overall, the article it's well written and clear.Weaknesses:Many of the experiments confirmed previous published data, which also show a decline of CERS1 in ageing and the generation and characterization of a muscle specific knockout mouse line. The mechanistic insights of how the increased amount of long ceramides (cer c24) and the decreased of shorter ones (cer c18) might influence muscle mass, force production, fibrosis and inflammation in aged mice have not been addressed.

We thank the reviewer for the assessment and would like to point out that Cers1 had not previously been studied in the context of aging. Moreover, our unbiased pathway analyses in human skeletal muscle implicate CERS1 for the first time with myogenic differentiation, which we validate in cell culture systems. To improve mechanistic insights, as suggested by Reviewer #1, we performed more experiments to gain insights how Cers1 derived c18, and Cers2 derived c24 ceramide species affect myogenesis. We recently showed that knocking out Cers2 reduces c24:0/c24:1 and promotes muscle cell maturation (PMID: 37118545, Fig. 6m-r and Supplementary Fig. 5e). This suggests that the very long chain ceramides c24 might indeed be driving the effect we see upon Cers1 inhibition because we observe an accumulation of c24 ceramides upon Cers1 (c18) inhibition (Fig 2B, Fig 3B, Fig 4A, Fig S3E), which is associated with impaired muscle maturation (Fig 4B-C, Fig S3G-I, Fig S4G-I). To study whether impaired muscle cell differentiation upon Cers1 inhibition is dependent on Cers2, we knocked-down Cers1 alone, or in combination with the knockdown of Cers2. Results show that reduced muscle cell maturation mediated by Cers1KD is rescued by the simultaneous knockdown of Cers2 as shown by gene expression analyses and immunohistochemical validation and quantification. Hence, we believe that reducing Cers1 function during aging might lead to an increase in sphingosine levels as has been shown previously (PMID: 31692231). Increased sphingosine triggers cell apoptosis due to its toxicity (PMID: 12531554). Therefore, channeling accumulating sphingosine towards C24 ceramides may avoid toxicity but, as we show in this manuscript, will reduce the myogenic potential in muscle. However, if also C24 production is blocked by Cers2 inhibition, sphingosine is forced towards the production of other, potentially less toxic or myogenesis-impairing ceramides. We added these new data to the revised manuscript as new Fig 5D-E and new Fig S5G-I.

**Reviewer #2 (Public Review):**
Summary:The manuscript by Wohlwend et al. investigates the implications of inhibiting ceramide synthase Cers1 on skeletal muscle function during aging. The authors propose a role for Cers1 in muscle myogenesis and aging sarcopenia. Both pharmacological and AAV-driven genetic inhibition of Cers1 in 18month-old mice lead to reduced C18 ceramides in skeletal muscle, exacerbating age-dependent features such as muscle atrophy, fibrosis, and center-nucleated fibers. Similarly, inhibition of the Cers1 orthologue in *C. elegans* reduces motility and causes alterations in muscle morphology.Strengths:The study is well-designed, carefully executed, and provides highly informative and novel findings that are relevant to the field.Weaknesses:The following points should be addressed to support the conclusions of the manuscript.(1) It would be essential to investigate whether P053 treatment of young mice induces age-dependent features besides muscle loss, such as muscle fibrosis or regeneration. This would help determine whether the exacerbation of age-dependent features solely depends on Cers1 inhibition or is associated with other factors related to age- dependent decline in cell function. Additionally, considering the reported role of Cers1 in whole-body adiposity, it is necessary to present data on mice body weight and fat mass in P053treated aged-mice.

We thank the reviewer to suggest that we study Cers1 inhibition in young mice. In fact, a previous study shows that muscle-specific Cers1 knockout in young mice impairs muscle function (PMID: 31692231). Similar to our observation, these authors report reduced muscle fiber size and muscle force. Therefore, we do not believe that our observed effects of Cers1 inhibition in aged mice are specific to aging, although the phenotypic consequences are accentuated in aged mice. As requested by the reviewer, we attached the mice body weights and fat mass (Author response image 1A-B). The reduced fat mass upon P053 treatment is in line with previously reported reductions in fat mass in chow diet or high fat diet fed young mice upon Cers1 inhibition (PMID: 30605666, PMID: 30131496), again suggesting that the effect of Cers1 inhibition might not be specific to aging.

**Author response image 1. sa3fig1:** (A-B) Body mass (A) and Fat mass as % of body mass (B) were measured in 22mo C57BL/6J mice intraperitoneally injected with DMSO or P053 using EchoMRI (n=7-12 per group). (C-D) Grip strengh measurements in all limbs (C) or only the forelimbs (D) in 24mo C57BL/6J mice intramuscularly injected with AAV9 particles containing scramble, or shRNA targeting Cers1 (n=8 per group). (E-F) Pax7 gene expression in P053 or AAV9 treated mice (n=6-7 per group) (E), or in mouse C2C12 muscle progenitor cells treated with 25nM scramble or Cers1 targeting shRNA (n=8 per group) (F). (G) Proliferation as measured by luciferase intensity in mouse C2C12 muscle muscle cells treated with 25nM scramble or Cers1 targeting shRNA (n=24 per group). Each column represents one biological replicate. (H) Overlayed FACS traces of Annexin-V (BB515, left) and Propidium Iodide (Cy5, right) of mouse C2C12 muscle myotubes treated with 25nM scramble or Cers1 targeting shRNA (n=3 per group). Quantification right: early apoptosis (Annexin+-PI-), late apoptosis (Annexin+-PI+), necrosis (Annexin--PI+), viability (Annexin--PI-). (I) Normalized Cers2 gene expression in mouse C2C12 muscle muscle cells treated with 25nM scramble or Cers1 targeting shRNA (n=6-7 per group). (J-K) Representative mitochondrial respiration traces of digitonin-permeablized mouse C2C12 muscle muscle cells treated DMSO or P053 (J) with quantification of basal, ATP-linked, proton leak respiration as well as spare capacity and maximal capacity linked respiration (n=4 per group). (L) Reactive oxygen production in mitochondria of mouse C2C12 muscle muscle cells treated DMSO or P053. (M) Enriched gene sets related to autophagy and mitophagy in 24mo C57BL/6J mouse muscles intramuscularly injected with AAV9 particles containing scramble, or shRNA targeting Cers1 (left), or intraperitoneally injected with DMSO or P053 (right). Color gradient indicates normalized effect size. Dot size indicates statistical significance (n=6-8 per group). (N) Representative confocal Proteostat stainings with quantifications of DMSO and P053 treated mouse muscle cells expressing APPSWE (top) and human primary myoblasts isolated from patients with inclusion body myositis (bottom). (O) Stillness duration during a 90 seconds interval in adult day 5 C. elegans treated with DMSO or 100uM P053. (P) Lifespan of *C. elegans* treated with DMSO or P053. (n=144-147 per group, for method details see main manuscript page 10).

(2) As grip and exercise performance tests evaluate muscle function across several muscles, it is not evident how intramuscular AAV-mediated Cers1 inhibition solely in the gastrocnemius muscle can have a systemic effect or impact different muscles. This point requires clarification.

The grip strength measurements presented in the manuscript come from hindlimb grip strength, as pointed out in the Methods section. We measured grip strength in all four limbs, as well as only fore- (Author response image 1C-D). While forelimb strength did not change, only hindlimb grip strength was significantly different in AAV-Cers1KD compared to the scramble control AAV (Fig 3I), which is in line with the fact that we only injected the AAV in the hindlimbs. This is similar to the effect we observed with our previous data where we saw altered muscle function upon IM AAV delivery in the gastrocnemius (PMID: PMID: 34878822, PMID: 37118545). The gastrocnemius likely has the largest contribution to hindlimb grip strength given its size, and possibly even overall grip strength as suggested by a trend of reduced grip strength in all four limbs (Author response image 1C). We also suspect that the hindlimb muscles have the largest contribution to uphill running as we could also see an effect on running performance. While we carefully injected a minimal amount of AAV into gastrocnemius to avoid leakage, we cannot completely rule out that some AAV might have spread to other muscles. We added this information to the discussion of the manuscript as a potential limitation of the study.

(3) To further substantiate the role of Cers1 in myogenesis, it would be crucial to investigate the consequences of Cers1 inhibition under conditions of muscle damage, such as cardiotoxin treatment or eccentric exercise.While it would be interesting to study Cers1 in the context of muscle regeneration, and possibly mouse models of muscular dystrophy, we think such work would go beyond the scope of the current manuscript.(4) It would be informative to determine whether the muscle defects are primarily dependent on the reduction of C18-ceramides or the compensatory increase of C24-ceramides or C24-dihydroceramides.

To improve mechanistic insights, as suggested by Reviewer #2, we performed more experiments to gain insights how Cers1 derived c18, and Cers2 derived c24 ceramide species affect myogenesis. We recently showed that knocking out Cers2 reduces c24:0/c24:1 and promotes muscle cell maturation (PMID: 37118545, Fig. 6m-r and Supplementary Fig. 5e). This suggests that the very long chain ceramides c24 might indeed be driving the effect we see upon Cers1 inhibition because we observe an accumulation of c24 ceramides upon Cers1 (c18) inhibition (Fig 2B, Fig 3B, Fig 4A, Fig S3E), which is associated with impaired muscle maturation (Fig 4B-C, Fig S3G-I, Fig S4G-I). To study whether impaired muscle cell differentiation upon Cers1 inhibition is dependent on Cers2, we knocked-down Cers1 alone, or in combination with the knockdown of Cers2. Results show that reduced muscle cell maturation mediated by Cers1KD is rescued by the simultaneous knockdown of Cers2 as shown by gene expression analyses and immunohistochemical validation and quantification. We added these data to the manuscript as new Fig 5D-E, new Fig S5G-I. These data, together with our previous results showing that Degs1 knockout reduces myogenesis (PMID: 37118545, Fig. 6s-x and Fig. 7) suggest that C24/dhC24 might contribute to the age-related impairments in myogenesis. We added the new results to the revised manuscript.

(5) Previous studies from the research group (PMID 37118545) have shown that inhibiting the de novo sphingolipid pathway by blocking SPLC1-3 with myriocin counteracts muscle loss and that C18-ceramides increase during aging. In light of the current findings, certain issues need clarification and discussion. For instance, how would myriocin treatment, which reduces Cers1 activity because of the upstream inhibition of the pathway, have a positive effect on muscle? Additionally, it is essential to explain the association between the reduction of Cers1 gene expression with aging (Fig. 1B) and the age-dependent increase in C18-ceramides (PMID 37118545).

Blocking the upstream enzyme of the ceramide pathway (SPT1) shuts down the entire pathway that is overactive in aging, and therefore seems beneficial for muscle aging. While most enzymes in the ceramide pathway that we studied so far (SPTLC1, CERS2) revealed muscle benefits in terms of myogenesis, inflammation (PMID: 35089797; PMID: 37118545) and muscle protein aggregation (PMID: 37196064), the CERS1 enzyme shows opposite effects. This is also visible in the direction of CERS1 expression compared to the other enzymes in one of our previous published studies (PMID: 37118545, Fig. 1e and Fig. 1f). In the current study, we show that Cers1 inhibition indeed exacerbates age-related myogenesis and inflammation as opposed to the inhibition of Sptlc1 or Cers2. As the reviewer points out, both C18- and C24-ceramides seem to accumulate upon muscle aging. We think this is due to an overall overactive ceramide biosynthesis pathway. Blocking C18-ceramides via Cers1 inhibition results in the accumulates C24-ceramides and worsens muscle phenotypes (see reply to question #4). On the other hand, blocking C24-ceramides via Cers2 inhibition improves muscle differentiation. These observations together with the finding that Cers1 mediated inhibition of muscle differentiation is dependent on proper Cers2 function (new Fig 5D-E, new Fig S5G-I) points towards C24-ceramides as the main culprit of reduced muscle differentiation. Hence, at least a significant part of the benefits of blocking SPTLC1 might have been related to reducing very long-chain ceramides. We believe that reduced Cers1 expression in skeletal muscle upon aging, observed by us and others (PMID: 31692231), might reflect a compensatory mechanism to make up for an overall overactive ceramide flux in aged muscles. Reducing Cers1 function during aging might lead to an increase in sphingosine levels as has been shown previously (PMID: 31692231). Increased sphingosine triggers cell apoptosis due to its toxicity (PMID: 12531554). Therefore, channeling accumulating sphingosine towards C24 ceramides may avoid toxicity but, as we show in this manuscript, will reduce the myogenic potential in muscle. However, if also C24 production is blocked by Cers2 inhibition (new Fig 5E-D, new Fig S5G-I), sphingosine is forced towards the production of other, potentially less toxic, or myogenesis-impairing ceramides. These data are now added to the revised manuscript (see page 7). Details were added to the discussion of the manuscript (see page 8).

Addressing these points will strengthen the manuscript's conclusions and provide a more comprehensive understanding of the role of Cers1 in skeletal muscle function during aging.

**Reviewer #1 (Recommendations For The Authors):**
The authors identified that genetical and pharmacological inhibition of CERS1, an enzyme implicated in ceramides biosynthesis worsen muscle fibrosis and inflammation during aging.Even though many of the experiments only confirmed previous published data (ref 21, 11,37,38), which also show a decline of CERS1 in ageing and the generation and characterization of a muscle specific knockout mouse line, the study points out an interesting issue on excluding CERS1 inhibition as a therapeutic strategy for sarcopenia and opens new questions on understanding how inhibition of SPTLC1 (upstream CERS1) have beneficial effects in healthy aging (ref 15 published by the same authors).Overall, the article it's well written and clear. However, there is a major weakness. The mechanistic insights of how the increased amount of long ceramides (c24) and the decreased of shorter ones (cer c18) might influence muscle mass, force production, fibrosis and inflammation in aged mice have not been addressed. At the present stage the manuscript is descriptive and confirmatory of CERS1 mediated function in preserving muscle mass. The authors should consider the following points:Comments:(1) Muscle data(a) The effect of CERS1 inhibition on myotube formation must be better characterized. Which step of myogenesis is affected? Is stem cell renewal or MyoD replication/differentiation, or myoblast fusion or an increased cell death the major culprit of the small myotubes? Minor point: Figure S1C: show C14:00 level at 200 h; text of Fig S2A and 1F: MRF4 and Myogenin are not an early gene in myogenesis please correct, Fig S2B and 2C: changes in transcript does not mean changes in protein or myotube differentiation and therefore, authors must test myotube formation and myosin expression.

Cers1 inhibition seems to affect differentiation and myoblast fusion. To test other suggested effects we performed more experiments as delineated. Inhibiting Cers1 systemically with the pharmacological inhibitor of Cers1 (P053) or with intramuscular delivery of AAV expressing a short hairpin RNA (shRNA) against Cers1 in mice did not affect Pax7 transcript levels (Author response image 1E). Moreover, we did also not observe an effect of shRNA targeting Cers1 on Pax7 levels in mouse C2C12 muscle progenitor cells (Author response image 1F). To characterize the effect of Cers1 inhibition on muscle progenitor proliferation/renewal, we used scramble shRNA, or shRNA targeting Cers1 in C2C12 muscle progenitors and measured proliferation using CellTiter-Glo (Promega). Results showed that Cers1KD had no significant effect on cell proliferation (Author response image 1G). Next, we assayed cell death in differentiating C2C12 myotubes deficient in Cers1 using FACS Analysis of Annexin V (left) and propidium iodide (right). We found no difference in early apoptosis, late apoptosis, necrosis, or muscle cell viability, suggesting that cell death can be ruled out to explain smaller myotubes (Author response image 1H). These findings support the notion that the inhibitory effect of Cers1 knockdown on muscle maturation are primarily based on effects on myogenesis rather than on apoptosis. Our data in the manuscript also suggests that Cers1 inhibition affects myoblast fusion, as shown by reduced myonucleation upon Cers1KD (Fig S3H right, Fig S5I).

(b) The phenotype of CESR1 knockdown is milder than 0P53 treated mice (Fig S5D and Figure 3F, 3H are not significant) despite similar changes of Cer18:0, Cer24:0, Cer 24:1 concentration in muscles . Why?

Increases in very long chain ceramides were in fact larger upon P053 administration compared to AAVmediated knockdown. For example, Cer24:0 levels increased by >50% upon P053 administration, compared to 20% by AAV injections. Moreover, dhC24:1 increased by 6.5-fold vs 2.5-fold upon P053 vs AAV treatment, respectively. These differences might not only explain the slightly attenuated phenotypes in the AA- treated mice but also underlines the notion that very long chain ceramides might cause muscle deterioration. We believe inhibiting the enzymatic activity of Cers1 (P053) as compared to degrading Cers1 transcripts is a more efficient strategy to reduce ceramide levels. However, we cannot completely rule out multi-organ, systemic effects of P053 treatment beyond its direct effect on muscle. We added these details in the discussion of the revised manuscript (see page 8 of the revised manuscript).

(c) The authors talk about a possible compensation of CERS2 isoform but they never showed mRNA expression levels or CERS2 protein levels aner treatment. Is CERS2 higher expressed when CERS1 is downregulated in skeletal muscle?

We appreciate the suggestion of the reviewer. We found no change in Cers2 mRNA levels upon Cers1 inhibition in mouse C2C12 myoblasts (Author response image 1I). We would like to point out that mRNA abundance might not be the optimal measurement for enzymes due to enzymatic activities. Therefore, we think metabolite levels are a better proxy of enzymatic activity. It should also be pointed out that “compensation” might not be an accurate description as sphingoid base substrate might simply be more available upon Cers1KD and hence, more substrate might be present for Cers2 to synthesize very long chain ceramides. This “re-routing” has been previously described in the literature and hypothesized to be related to avoid toxic (dh)sphingosine accumulation (PMID: 30131496). Therefore, we changed the wording in the revised manuscript to be more precise.

(d) Force measurement of AAV CERS1 downregulated muscles could be a plus for the study (assay function of contractility)

In the current study we measured grip strength in mice, which had previously been shown to be a good proxy of muscle strength and general health (PMID: 31631989). Indeed, our results of reduced muscle grip strength are in line with previous work that shows reduced contractility in muscles of Cers1 deficient mice (PMID: 31692231).

(e) How are degradation pathways affected by the downregulation of CERS1. Is autophagy/mitophagy affected? How is mTOR and protein synthesis affected? There is a recent paper that showed that CerS1 silencing leads to a reduction in C18:0-Cer content, with a subsequent increase in the activity of the insulin pathway, and an improvement in skeletal muscle glucose uptake. Could be possible that CERS1 downregulation increases mTOR signalling and decreases autophagy pathway? Autophagic flux using colchicine in vivo would be useful to answer this hypothesis

Cers1 in skeletal muscle has indeed been linked to metabolic homeostasis (see PMID: 30605666). In line with their finding in young mice we also find reduced fat mass upon P053 treatment in aged mice (Author response image 1A-B). We also looked into mitochondrial bioenergetics upon blocking Cers1 with P053 treatment using an O2k oxygraphy (Author response image 1J-L). Results show that Cers1 inhibition in mouse muscle cells increases mitochondrial respiration, similar to what has been shown before (PMID: 30131496). However, we also found that reactive oxygen species production in mouse muscle cells is increased upon P053 treatment, suggesting the presence of dysfunctional mitochondria upon inhibiting Cers1 with P053.We next looked into the mitophagy/autophagy degradation pathways suggested by the reviewer and do not find convincing evidence supporting that Cers1 has a major impact on autophagy or mitophagy derived gene sets in mice treated with shRNA against Cers1, or the Cers1 pharmacological inhibitor P053 (Author response image 1M).

We then assessed the effect of Cers1 inhibition on transcripts levels related to the mTORC1/protein synthesis, as suggested by the reviewer. Cers1 knockdown in differentiating mouse muscle cells showed only a weak trend to reduce mTORC1 and its downstream targets (new Fig S4A). In line with this, there was no notable difference in protein synthesis in differentiating, Cers1 deficient mouse C2C12 myoblasts as assessed by L-homopropargylglycine (HPG) amino acid labeling using confocal microscopy (new Fig S4B) or FACS analyses (new Fig S4C). However, Cers1KD increased transcripts related to the myostatin-Foxo1 axis as well as the ubiquitin proteasome system (e.g. atrogin-1, MuRF1) (new Fig S4D), suggesting Cers1 inhibition increases protein degradation. We added these details to the revised manuscript on page 7. We recently implicated the ceramide pathway in regulating muscle protein homeostasis (PMID: 37196064). Therefore, we assessed the effect of Cers1 inhibition with the P053 pharmacological inhibitor on protein folding in muscle cells using the Proteostat dye that intercalates into the cross-beta spine of quaternary protein structures typically found in misfolded and aggregated proteins. Interestingly, inhibiting Cers1 further increased misfolded proteins in C2C12 mouse myoblasts expressing the Swedish mutation in APP and human myoblasts isolated from patients with inclusion body myositis (Author response imageure 1N). These findings suggest that deficient Cers1 might upregulate protein degradation to compensate for the accumulation of misfolded and aggregating proteins, which might contribute to impaired muscle function observed upon Cers1 knockdown. Further studies are needed to disentangle the underlying mechanstics.

(f) The balances of ceramides have been found to play roles in mitophagy and fission with an impact on cell fate and metabolism. Did the authors check how are mitochondria morphology, mitophagy or how dynamics of mitochondria are altered in CERS1 knockdown muscles? (fission and fusion). There is growing evidence relating mitochondrial dysfunction to the contribution of the development of fibrosis and inflammation.

Previously, CERS1 has been studied in the context of metabolism and mitochondria (for reference, please see PMID: 26739815, PMID: 29415895, PMID: 30605666, PMID: 30131496). In summary, these studies demonstrate that C18 ceramide levels are inversely related to insulin sensitivity in muscle and mitochondria, and that Cers1 inhibition improves insulin-stimulated suppression of hepatic glucose production and reduced high-fat diet induced adiposity. Moreover, improved mitochondrial respiration, citrate synthase activity and increased energy expenditure were reported upon Cers1 inhibition. Lack of Cers1 specifically in skeletal muscle was also reported to improve systemic glucose homeostasis. While these studies agree on the effect of Cers1 inhibition on fat loss, results on glucose homeostasis and insulin sensitivity differ depending on whether a pharmacologic or a genetic approach was used to inhibit Cers1. The current manuscript describes the effect of CERS1 on muscle function and myogenesis because these were the most strongly correlated pathways with CERS1 in human skeletal muscle (Fig 1C) and impact of Cers1 on these pathways is poorly studied, particularly in the context of aging. Therefore, we would like to refer to the mentioned studies investigating the effect of CERS1 on mitochondria and metabolism.

(2) *C. elegans* data:(a) The authors checked maternal RNAi protocol to knockdown lagr-1 and showed alteration of muscle morphology at day 5. They also give pharmacological exposure of P053 drug at L4 stage. Furthermore, the authors also used a transgenic ortholog lagr-1 to perform the experiments. All of them were consistent showing a reduced movement. It would be important to show rescue of the muscle phenotype by overexpressing CERS1 ortholog in knockdown transgenic animals.

We used RNAi to knockdown the Cers1 orthologue, lagr-1, in *C. elegans*. Therefore, we do not have transgenic animals. Overexpressing lagr-1 in the RNAi treated animals would also not be possible as the RNA from the overexpression would just get degraded.

(b) The authors showed data about distance of *C. elegans*. It would be interesting to specify if body bends, reversals and stillness are affected in RNAi and transgenic Knockdown worms.

As suggested, we measured trashing and stillness as suggested by the reviewer and found reduced trashing (new Fig S5B) and a trend towards an increase in stillness (Author response image 1O) in P053 treated worms on day 5 of adulthood, which is the day we observed significant differences in muscle morphology and movement (Fig 4D-E, Fig S5A). These data are now included in the revised manuscript.

(c) Is there an effect on lifespan extension by knocking down CERS1?

We performed two independent lifespan experiments in *C. elegans* treated with the Cers1 inhibitor P053 and found reduced lifespan in both replicate experiments (for second replicate, see Author response image 1P). We added these data to the revised manuscript as new Fig 4H.

How do the authors explain the beneficial effect of sptlc1 inhibition on healthy aging muscle? Discuss more during the article if there is no possible explanation at the moment.

We believe that blocking the upstream enzyme of the ceramide pathway (SPT1) shuts down the entire pathway that is overactive in aging, and therefore is more beneficial for muscle aging. Our current work suggests that at least a significant part of Sptlc1-KD benefits might stem from blocking very long chain ceramides. While SPTLC1 and CERS2 revealed muscle benefits in terms of myogenesis, inflammation (PMID: 35089797; PMID: 37118545) and muscle protein aggregation (PMID: 37196064), the CERS1 enzyme shows opposite effects, which is also visible in Fig 1e and Fig 1f of PMID: 37118545. In the current study, we show that Cers1 inhibition indeed exacerbates aging defects in myogenesis and inflammation as opposed to the inhibition of Sptlc1 or Cers2. The fact that the effect of Cers1 on inhibiting muscle differentiation is dependent on the clearance of Cers2-derived C24-ceramides suggests that reducing very long chain ceramides might be crucial for healthy muscle aging. We added details to the discussion.